# Epsin and Sla2 form assemblies through phospholipid interfaces

Maria M. Garcia-Alai[1], Johannes Heidemann[2], Michal Skruzny[3], Anna Gieras[1,4], Haydyn D.T. Mertens[1], Dmitri I. Svergun[1], Marko Kaksonen[5], Charlotte Uetrecht [2,6] & Rob Meijers [1]

In clathrin-mediated endocytosis, adapter proteins assemble together with clathrin through interactions with specific lipids on the plasma membrane. However, the precise mechanism of adapter protein assembly at the cell membrane is still unknown. Here, we show that the membrane–proximal domains ENTH of epsin and ANTH of Sla2 form complexes through phosphatidylinositol 4,5-bisphosphate (PIP2) lipid interfaces. Native mass spectrometry reveals how ENTH and ANTH domains form assemblies by sharing PIP2 molecules. Furthermore, crystal structures of epsin Ent2 ENTH domain from *S. cerevisiae* in complex with PIP2 and Sla2 ANTH domain from *C. thermophilum* illustrate how allosteric phospholipid binding occurs. A comparison with human ENTH and ANTH domains reveal only the human ENTH domain can form a stable hexameric core in presence of PIP2, which could explain functional differences between fungal and human epsins. We propose a general phospholipid-driven multifaceted assembly mechanism tolerating different adapter protein compositions to induce endocytosis.

[1] European Molecular Biology Laboratory (EMBL), Hamburg Outstation, Notkestrasse 85, 22607 Hamburg, Germany. [2] Heinrich Pette Institute Leibniz Institute for Experimental Virology, Martinistrasse 52, 20251 Hamburg, Germany. [3] LOEWE Center for Synthetic Microbiology (SYNMIKRO) Max Planck Institute for Terrestrial Microbiology, 35043 Marburg, Germany. [4] University Medical Center Hamburg – Eppendorf, Martinistrasse 52, 20246 Hamburg, Germany. [5] Department of Biochemistry and NCCR Chemical Biology, University of Geneva, Quai Ernest-Ansermet 30, 1211 Geneva 4, Switzerland. [6] European XFEL GmbH, Holzkoppel 4, 22869 Schenefeld, Germany. Maria M. Garcia-Alai and Johannes Heidemann contributed equally to this work. Correspondence and requests for materials should be addressed to C.U. (email: charlotte.uetrecht@xfel.eu) or to R.M. (email: r.meijers@embl-hamburg.de)

Clathrin-mediated endocytosis is essential for protein retrieval during neurotransmission and receptor recycling. It is also involved in viral entry and the uptake of nutrients and hormones. During endocytosis, a cargo-containing vesicle is formed through the invagination of a plasma membrane patch, in a process that involves proteins and specific lipids. Accumulation of the adapter proteins at the plasma membrane initiates the endocytic event and contributes to membrane bending[1]. The adapters recruit clathrin and associate with the actin cytoskeleton to accomplish vesicle budding[2]. Finally, the vesicle coated with clathrin and adapter proteins is detached from the plasma membrane by dynamin[3].

The size and shape of the vesicle is determined by the interplay of adapter proteins and clathrin[4]. Clathrin does not bind to the plasma membrane directly, but to a range of adapter proteins that interact with lipids on the plasma membrane. Many adapter proteins, such as epsin[5], AP180[6], and AP-2[7], contain positively charged patches that bind to the head groups of specific phospholipids. This binding is strong enough so that the adapter proteins remain attached to the membrane, and can undergo conformational changes to bind cargo[7]. Adapter proteins containing an epsin N-terminal homology (ENTH) domain[5,8] and some adapter proteins containing an AP180 N-terminal homology (ANTH) domain that belong to the subfamily of clathrin assembly lymphoid myeloid leukemia (CALM) proteins[9], have an additional membrane-binding mode, inserting an N-terminal helix into the plasma membrane. It is proposed that there is a concerted mechanism where binding to the head group of phosphatidylinositol 4,5-bisphosphate (PIP2) triggers the N-terminal portion of these adapter proteins to fold into an α helix. The insertion of the α helix into the membrane contributes to its curvature[1].

It is unclear whether the clathrin-associated adapter proteins engage clathrin individually at the cell membrane, or whether they assemble in larger structures to drive membrane curvature and clathrin recruitment[10]. It has also been proposed that membrane curvature is caused simply by a protein crowding mechanism, where the formation of the clathrin coat crowds adapter proteins together, leading to membrane curvature[11]. Critically, the combination of membrane insertions and spherical scaffolds that curve the membrane have been observed[10], but their interactions are not understood. There are indications that adapter proteins themselves form larger oligomers, preceding the formation of the clathrin coat. Complex formation has been observed between the ENTH domain of epsin and the ANTH domain of Sla2, the yeast homolog of human huntingtin interacting protein 1 related (Hip1R)[12]. The formation of this complex depends on the presence of PIP2 and has been observed in *Saccharomyces cerevisiae*. A pulldown experiment with full-length epsin-1 and Hip1R from *Mus musculus* suggested the same complex forms in mammals[13]. Endocytosis is stalled in *S. cerevisiae* when the complex between the ENTH domain of epsin Ent1 and the ANTH domain of Sla2 is disrupted by mutagenesis[12]. Electron cryo-tomography studies on helical assemblies of epsin Ent1 ENTH and Sla2 ANTH domains obtained from giant unilamellar vesicles (GUV)s revealed that the co-assembly of ENTH/ANTH occurs[14]. The ENTH/ANTH assembly may help organize the clathrin coat and could contribute to membrane curvature during the initial stages of endocytosis[14]. It is of particular interest that epsin and Sla2 form a membrane-bound scaffold, because these adapter proteins have also been associated with actin recruitment to the coated pit. Actin polymerization is known to provide mechanical force for vesicle formation from membranes under tension[15]. The adapter assembly would harness singular weak interactions together to form an anchor that can withstand the strong forces generated by the actin cytoskeleton.

In this paper, we reveal the assembly process of epsin ENTH and Sla2/Hip1R ANTH domains through PIP2 interfaces by native MS. An X-ray crystal structure of the ENTH domain of epsin Ent2 from *S. cerevisiae* in complex with PIP2 reveals an allosteric mechanism underlying the assembly. The crystal structure of the Sla2 ANTH domain from *C. thermophilum* reveals how this subfamily of ANTH domains evolved to engage ENTH domains in assembly. We determine by native MS that some elements of the assembly process are evolutionarily conserved between the fungal and human adapter proteins. However, human epsin ENTH can autonomously form homo-oligomers, whereas fungal ENTH domains need the presence of Sla2 ANTH domains to create stable assemblies. Together, these data suggest that the ENTH/ANTH complex concentrates PIP2 locally on the plasma membrane to facilitate the formation of adapter complexes with cargo, clathrin, or other adapter proteins through PIP2-dependent interfaces.

## Results

**Cooperative binding of PIP2 to the ENTH domain of epsin.** To investigate how a lipid-dependent protein complex is formed between epsin ENTH domain and the Sla2/Hip1R ANTH domain, we first determined the binding between the PIP2 lipid (diC8-PIP2) and ENTH domains from *S. cerevisiae* alone using native MS[16,17]. The Ent1 and Ent2 ENTH domains (ENTH1 and ENTH2) from *S. cerevisiae* as well as an ENTH domain of homologous epsin from *C. thermophilum* bind up to two PIP2 molecules per domain (Fig. 1a, b). Based on a direct MS approach[18,19], macroscopic dissociation constants were determined revealing dependence of the two PIP2-binding sites by positive cooperativity[20] (Fig. 1c and Supplementary 11). Results were comparable for the MS optimized ammonium acetate concentration of 300 mM and the more physiological ionic strength of 160 mM ammonium acetate (Supplementary Table 1).

A crystal structure was determined of a complex between the ENTH domain of Ent2 (ENTH2) from *S. cerevisiae* and the PIP2 phospholipid to a resolution of 3.4 Å (Table 1). The structure shows two ENTH2 domains sandwiched around one PIP2 molecule (Fig. 2a and Supplementary Fig. 1). The inositol head group of PIP2 is bound to the previously identified phosphatidylinositol-binding pocket of one of the ENTH2 molecules[5]. The N-terminal region of this ENTH2 molecule is folded into an α helix that corresponds to the helix (α0) that has been observed in the ENTH domain of rat epsin-1 in complex with the phosphoinositol (IP3) group[5], and this copy is labeled ENTH2$_{\alpha 0}$. In the second ENTH2 molecule, the N-terminal α0 is unfolded (ENTH2$_{No\alpha 0}$), and the electron density suggests the N-terminal residues are extended. A superimposition of the ENTH1/IP3 domain complex structure of rat epsin-1[5] onto the ENTH2/PIP2 domain crystal structure from *S. cerevisiae* results in an RMSD of 1.1 Å for 147 residues showing distinct orientations of the α0 helix (Fig. 2b). In the ENTH1/IP3 structure, the α0 helix folds back onto the phosphoinositol group, so that Arg 8 and Lys 11 interact with an inositol head group. In the ENTH2/PIP2 structure, the α0 helix points away from the ENTH2 domain and there are no interactions with the phosphoinositol group. Instead, the evolutionarily conserved residue Tyr 16 of the ENTH2$_{\alpha 0}$ molecule forms direct interactions with the inositol head group of PIP2.

As a result of α0 reorientation pointing away from the ENTH2$_{\alpha 0}$ molecule, the second ENTH2$_{No\alpha 0}$ molecule can form interactions with the phosphatidylinositol head group. Interestingly, the ENTH2$_{No\alpha 0}$ domain is oriented toward the PIP2

molecule to involve residue Thr 104 on α6 in the ENTH2 dimer interface. This residue is crucial for the formation of the ENTH/ANTH complex[12,14,21]. Thr 104 of ENTH2$_{No\alpha0}$ is in the periphery of the PIP2-binding site, and sits opposite Tyr 68 and Lys 69 of the ENTH2$_{\alpha0}$ molecule. Thr 104 is conserved in epsin ENTH domains and is essential for ENTH function, though independent from the canonical PIP2-binding site[22,23]. Native MS on a Thr104Glu mutant of ENTH1 from *S. cerevisiae* shows a reduction of PIP2 binding, where the binding of two PIP2 molecules is barely observed (Fig. 1b), indicating impairment of the binding site.

To analyze the contribution of the PIP2 molecule to complex formation, we calculated the buried surface area contributions for each molecule with PISA[24]. The total solvent accessible area of the phosphatidylinositol head group of PIP2 is 488 Å$^2$. Most of the head group is buried by the ENTH2$_{\alpha0}$ molecule (306 Å$^2$, or 63% of the total available surface area of the PIP2 molecule),

whereas the ENTH2$_{No\alpha0}$ molecule covers 95 Å$^2$, or 20% of the available surface area. The buried solvent area between ENTH2$_{\alpha0}$ and ENTH2$_{No\alpha0}$ is 1820 Å$^2$, with almost equal contributions from ENTH2$_{\alpha0}$ (955 Å$^2$) and ENTH2$_{No\alpha0}$ (865 Å$^2$). However, most of the buried area (1020 Å$^2$) contributed to the dimer interface comes from the α0 helix of the ENTH$_{\alpha0}$ domain, which is repositioned by the PIP2 molecule to facilitate dimer formation. The effect of PIP2 binding is therefore twofold; it attaches the ENTH domain to the membrane and displaces the α0 helix, so it can form a dimer with a second ENTH domain. As a further indication that the ENTH2 dimer, which sandwiched PIP2, is functionally relevant, a similar dimer interface is observed in a crystal structure of the epsin Ent1 ENTH domain (ENTH1) from *S. cerevisiae* at 2.9 Å resolution (Fig. 2c). Here, a 2-methyl-2,4-pentanediol (MPD) molecule from the crystallization solution has caused a displacement of the α0 helix to allow dimer formation between ENTH1 domains.

In the ENTH2/PIP2 structure, two α0 helices from symmetry-related ENTH2$_{\alpha0}$ molecules stack against each other to create a homotetramer containing two PIP2 molecules in the same plane, ready to insert into the membrane (Fig. 2d). In this planar orientation, four ENTH molecules line up so they could associate with the plasma membrane, whereas the C-termini are pointing away from the membrane. The α0 helices of the ENTH2$_{\alpha0}$ domains are not positioned to insert into the membrane. However, the unfolded N-terminal regions of ENTH2$_{No\alpha0}$ as well as the neighboring positively charged PIP2-binding patch consisting of residues Lys 14, Arg 24, Arg 72, and His 72 are lying in the same plane as the lipid tail of PIP2 being aligned with the cell membrane. The N-terminus is therefore ideally positioned to bind another PIP2 molecule on the plasma membrane. We thus conclude that the PIP2/ENTH2 crystal structure shows an intermediate state, where the binding of one PIP2 dimerizes two ENTH2 domains and prepares the "empty" ENTH domain to bind another PIP2 molecule. This allosteric switch between PIP2 binding, dimerization, and the formation of α0 could potentially create larger epsin clusters on the cell membrane.

**The Sla2 ANTH domain evolved to bind the ENTH/PIP2 complex.** Native MS was also used to investigate PIP2 binding to

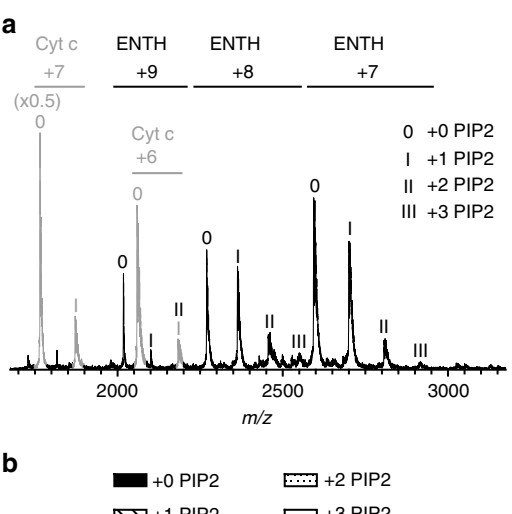

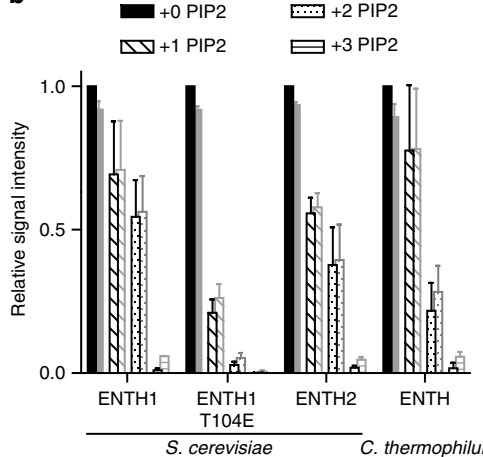

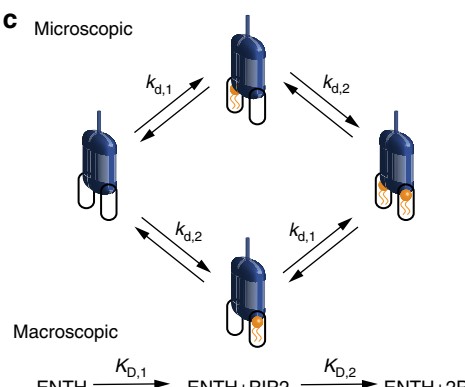

**Fig. 1** Native MS of ENTH/PIP2 complexes suggests an allosteric binding mechanism. **a** For analyzing lipid binding to ENTH domains, proteins were measured in presence of PIP2 and cytochrome *c* as reference. Raw spectra show free cytochrome *c* and unspecific attachment of 1 PIP2 (gray), while ENTH domains (here *C. thermophilum*) bind 0–3 PIP2. **b** Signal intensities from MS were summed over all charge states (back) and corrected for unspecific PIP2 clustering based on the ratio of bound/unbound reference protein (front). Data of at least three independent measurements were normalized to the corrected signal of unbound ENTH and the averages of the relative signal intensities and their standard deviations were plotted. The signal for ENTH with three PIP2 observed in raw spectra disappears after correction. **c** Schematic illustration of microscopic and macroscopic dissociation constants of two PIP2 molecules (orange) binding independently to ENTH (blue). For the first binding event of PIP2, two pathways with the microscopic dissociation constants $k_{d,1}$ and $k_{d,2}$ are available, leading to one apparent species of ENTH+PIP2. Combined, they account for the macroscopic dissociation constant $K_{D,1}$. The second macroscopic dissociation constant $K_{D,2}$ describes PIP2 binding to the thus far unoccupied binding site that yields the product ENTH+2PIP2. Again, this binding event can be partitioned into two pathways with the microscopic dissociation constants $k_{d,1}$ and $k_{d,2}$. If $k_{d,1}$ and $k_{d,2}$ are unaltered in the first and second binding event, binding sites are independent. Binding sites are represented by rounded rectangles

**Table 1 Data collection and refinement statistics**

|  | ENTH2 (S. cerevisiae) | ENTH2/PIP2 (S. cerevisiae) | ENTH1 (S. cerevisiae) | ANTH$_{Sla2}$ (C. thermophilum) |
|---|---|---|---|---|
| *Data collection* |  |  |  |  |
| Space group | $P2_1$ | $F432$ | $C222_1$ | $P2_1$ |
| Unit cell dimensions |  |  |  |  |
| *a, b, c* (Å) | 24.4, 65.7, 33.7 | a,b,c=211.4 | 68.9, 119.4, 130.0 | 47.8, 106.9, 51.6 |
| $\alpha, \beta, \gamma$ (°) | $\beta$=97.3 |  |  | $\beta$=98.0 |
| Resolution (Å) | 66–1.80 (1.84–1.980)[a] | 120–3.35 (8.85–3.35)[a] | 27–2.75 (2.90–2.75)[a] | 47–1.84 (1.88–1.84)[a] |
| $R_{pim}$ | 0.020 (0.353) | 0.053 (1.04) | 0.049 (0.49) | 0.062 (0.752) |
| $R_{merge}$ | 0.048 (0.580) | 0.134 (1.62) | 0.095 (0.80) | 0.12 (1.42) |
| $I/\sigma I$ | 23.6 (2.1) | 9.4 (0.9) | 11.8 (1.2) | 8.9 (1.0) |
| Completeness (%) | 99.2 (94.6) | 98.6 (97.1) | 99.1 (99.1) | 99.5 (93.6) |
| Redundancy | 6.4 (3.5) | 6.2 (3.0) | 5.3 (3.5) | 4.6 (4.4) |
| $CC_{1/2}$[a] | 0.76 | 0.34 | 0.71 | 0.35 |
| *Refinement* |  |  |  |  |
| Resolution (Å) | 33.5–1.90 | 75.0–3.35 (4.22–3.35) | 27.1–2.75 (2.82–2.75) | 47.0–1.84 (1.89–1.84) |
| No. reflections | 7571 | 5798 | 13379 | 41152 |
| $R_{work}/R_{free}$ | 0.17/0.22 | 0.28/0.30 | 0.24/0.29 | 0.18/0.23 |
| No. of atoms |  |  |  |  |
| Protein | 1104 | 2378 | 3578 | 5038 |
| Ligand/ion | N/A | 36 | 8 | 91 |
| Water | 61 | N/A | N/A | N/A |
| *B*-factors (Å$^2$) |  |  |  |  |
| Protein | 26 | 155 | 135 | 81 |
| Ligand/ion | N/A | 165 | 150 | 101 |
| Water | 35 | N/A | N/A | N/A |
| RMS deviations |  |  |  |  |
| Bond lengths (Å) | 0.018 | 0.004 | 0.015 | 0.017 |
| Bond angles (°) | 1.95 | 0.83 | 1.63 | 0.68 |
| Ramachandran statistics |  |  |  |  |
| Favored (%) | 100.0 | 91.9 | 91.4 | 91.5 |
| Disallowed (%) | 0.0 | 1.8 | 0.5 | 0.5 |

[a]Values in parentheses are for the highest-resolution shell

several ANTH domains, showing two PIP2 binding sites for all wild-type proteins (Fig. 3a and Supplementary Table 1). There is a striking difference in PIP2 binding between human CALM and the fungal Sla2 ANTH domains. The macroscopic dissociation constants for CALM suggest higher affinity for the first PIP2-binding site, and a lack of cooperative binding (Supplementary Table 1). Mutagenesis of the canonical PIP2-binding site on the Sla2 ANTH molecule from *S. cerevisiae* revealed the presence of a specific secondary binding site that has so far not been identified. Implementing the dissociation constant of the mutant into a mathematical model (Fig. 1c) revealed dependence and positive cooperativity of the two binding sites. In general, PIP2 dissociation constants are higher for Sla2 ANTH domains than for ENTH domains from *S. cerevisiae* showing $K_D$s of 100–250 μM (Supplementary Table 1).

To understand why the ANTH domain of Sla2/Hip1R is specifically suited to bind the ENTH domain of epsins through the lipid interface, we determined the crystal structure of the ANTH domain of homologous Sla2 from *C. thermophilum* to 1.8 Å resolution (Table 1). The overall ANTH fold consists of helices α1–α11 and is conserved between CALM and Sla2/Hip1R subfamilies (Fig. 3b). A structure-based sequence alignment of CALM subfamily members CALM and AP180[6] reveals two hitherto hidden, systematic and evolutionarily conserved features (Fig. 3b and Supplementary Fig. 2). Through insertion of a conserved Tyr 252 and a hydrophobic residue, an extra α helix (α12) uniquely occurs in Sla2 ANTH domains, providing rigidity. A loop connecting helices α8 and α9 has a five-residue extension in Sla2, although with non-conserved sequence. Based on the low-resolution electron microscopy tomogram of the Ent1 ENTH/ANTH Sla2 complex from *S. cerevisiae*, these insertions point at the epsin interface[14]. Moreover, deleting the Sla2/Hip1R-specific

α12 helix impaired the function of the Sla2 ANTH domain similarly to mutating Arg 29, which is essential for the ENTH/ANTH complex formation (Fig. 3c)[14]. In addition, the CALM ANTH domain has an N-terminal helix that is affecting membrane curvature[9], and that is similar to α0 observed in epsins. Based on the *C. thermophilum* Sla2 ANTH crystal structure, ANTH domains of the Sla2/Hip1R subfamily do not contain an N-terminal α helix. This is further confirmed by circular dichroism experiments performed on several ANTH and ENTH domains (Supplementary Fig. 3). Sla2/Hip1R ANTH domains display a loss of secondary structure or aggregation at high PIP2 concentrations (Supplementary Fig. 3c, f). To address whether this is caused by PIP2 itself, or the presence of an amphipathic environment, we repeated the circular dichroism measurements in the presence of submicellar and micellar concentrations of the detergent n-Dodecyl-beta-Maltoside (DDM) with and without PIP2 (Supplementary Fig. 3g, h). Human Hip1R ANTH domain shows a loss of signal with DDM (Supplementary Fig. 3h). The protein seems to undergo unfolding in the presence of an amphipathic environment, which would not be expected for a protein that contains a helix inducible by lipid binding. In contrast, CALM displays a clear increase in signal in the presence of DDM and PIP2 compared to the protein in buffer alone or with PIP2. In addition, when only the detergent above micellar concentrations is added, we further observe an increase in the signal. This experiment shows that it is actually the presence of the micellar environment and not only the phospholipid per se that is inducing/stabilizing helical structures in the protein and is in agreement with what has been previously reported for CALM that contains an α0 amphipathic helix[9]. We therefore conclude that N-terminal amphipathic helices are not

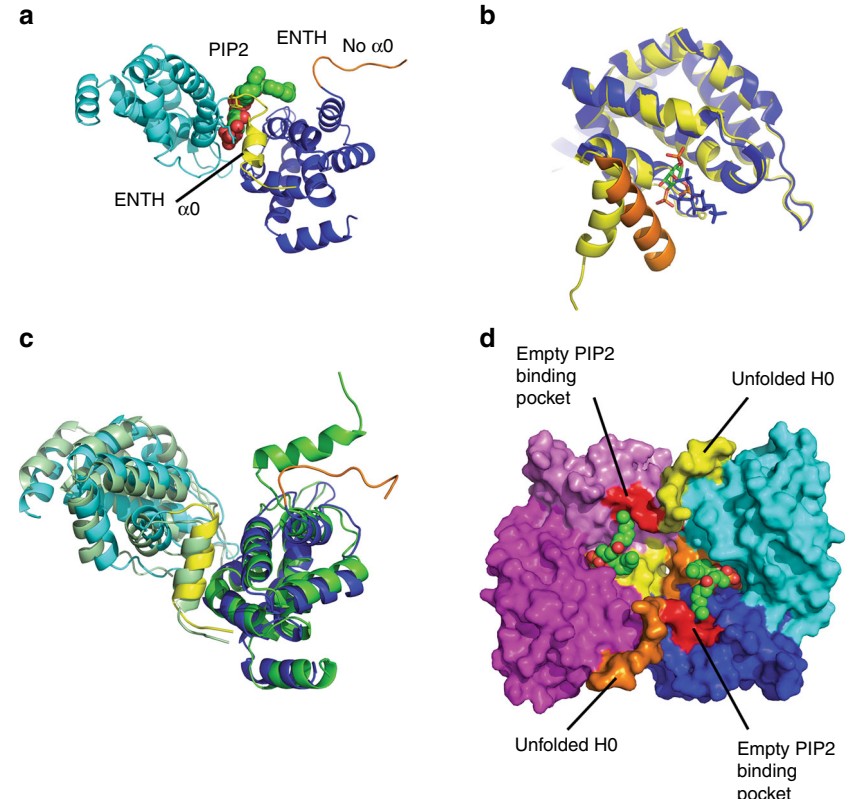

**Fig. 2** Crystal structure of the ENTH2/PIP2 complex reveals an allosteric-binding mechanism. **a** Ribbon diagram of the $ENTH2_{\alpha 0}$ (cyan) and $ENTH2_{No\alpha 0}$ dimer (blue) building block. N-terminal regions are colored ($ENTH2_{\alpha 0}$, yellow; $ENTH2_{No\alpha 0}$, orange). PIP2 sitting in the interface between ENTH2 molecules is shown as spheres. **b** Superposition of the ENTH/IP3 epsin-1 complex (dark blue) and the ENTH2/PIP2 complex (yellow). The N-terminal α0 is colored (ENTH2/PIP2, yellow; ENTH/IP3, orange), the inositol head groups are shown as sticks. **c** Superimposition of the ENTH2/PIP2 dimer (blue/cyan) and the ENTH1 dimer (green/palegreen) shown as a ribbon diagram. The α0 helix of ENTH1 is oriented similarly to α0 of $ENTH2_{\alpha 0}$. **d** Surface presentation of the ENTH2/PIP2 complex showing that unfolded N-termini of ENTH2 are in plane with the lipid tail of the PIP2 molecules (two $ENTH2_{\alpha 0}/ENTH2_{No\alpha 0}$ building blocks, cyan/blue and magenta/violet; PIP2, spheres; N-termini for empty ENTH2, yellow and orange; Tyr 16, Arg 24, Arg 62 and His 72 of $ENTH2_{No\alpha 0}$ with empty PIP2-binding pocket, red)

characteristic of either ANTH subfamily, but are only present in a subset of CALM adapter proteins.

**Ordered assembly formation of fungal ENTH and Sla2 ANTH.** To determine how universal PIP2-dependent interactions between ENTH and ANTH domains are, we analyzed PIP2-driven complex formation for a selection of ENTH and ANTH domains by native MS. We tested complex formation between the *S. cerevisiae* ENTH2 domain and the ANTH domain of Sla2 to verify the assembly relevant to the ENTH2/PIP2 crystal structure (Fig. 4). Indeed, an assembly with a stoichiometry of $8:8:24 \pm 3$ for $ENTH2/ANTH_{Sla2}/PIP2$ is formed, similar to the ENTH1 complex and $ENTH/ANTH_{Sla2}$ complex of *C. thermophilum* (Fig. 4 and Supplementary Table 2). When equal amounts of *S. cerevisiae* ENTH1 and ENTH2 are mixed together with $ANTH_{Sla2}$ and measured by native MS, a complex containing equal amounts of ENTH1 and ENTH2 is formed, showing no preference for either isoform and highlighting the possibility to recruit different epsins into a complex with Sla2 (Supplementary Fig. 4a). Furthermore, complex formation was impaired for the ENTH1 Thr 104 mutant (Supplementary Fig. 6), confirming previous ITC measurements[14].

Although the hetero 16-mer of the fungal $ENTH/ANTH_{Sla2}$ complex is most prominent, a smaller oligomer of ~330 kDa is also observed (Fig. 4) consisting of six ENTH and six ANTH domains ($ENTH1/2/ANTH_{Sla2}/PIP2$ is $6:6:19 \pm 2$). Further information on complex topology was obtained in collision-induced

dissociation (CID) MS/MS, in which proteins unfold partially until small proteins from the periphery are ejected. Here, hetero 12-mers and 16-mers show identical dissociation pathways. The larger ANTH domain is ejected from the complexes while all PIP2 and ENTH molecules remain bound to the complexes, indicating localization of ANTH in the periphery around an ENTH/PIP2 core (Fig. 3a and Supplementary Fig. 4).

It seems that the ENTH/ANTH/PIP2 mixtures form transient complexes that can alter in size. To understand the relationship between the two oligomeric states, oligomeric composition of a mixture of ENTH, $ANTH_{Sla2}$ (both *S. cerevisiae*), and PIP2 was monitored by native MS over time. The changing signal intensities revealed a maturation process enriching the larger hetero 16-mer (Fig. 5a). Amounts of hetero 12-mer and 16-mer complexes are approximately equal 1 min after mixing. Within 2 min, the 12-mer signal fades, and the 16-mer gets predominant. These results and the equilibrium ratio of 12-mers and 16-mers show that the hetero 16-mer is the more stable ENTH/ANTH/PIP2 complex, suggesting that the 12-mer is an assembly intermediate. Furthermore, complex assembly is reversible upon depletion and subsequent replenishing of PIP2 (Supplementary Fig. 4c).

**Human ENTH forms a hexameric core upon PIP2 binding.** It was previously reported that full-length Hip1R can only be pulled down by murine epsin ENTH in the presence of PIP2, suggesting that the PIP2-dependent formation of ENTH/ANTH complexes

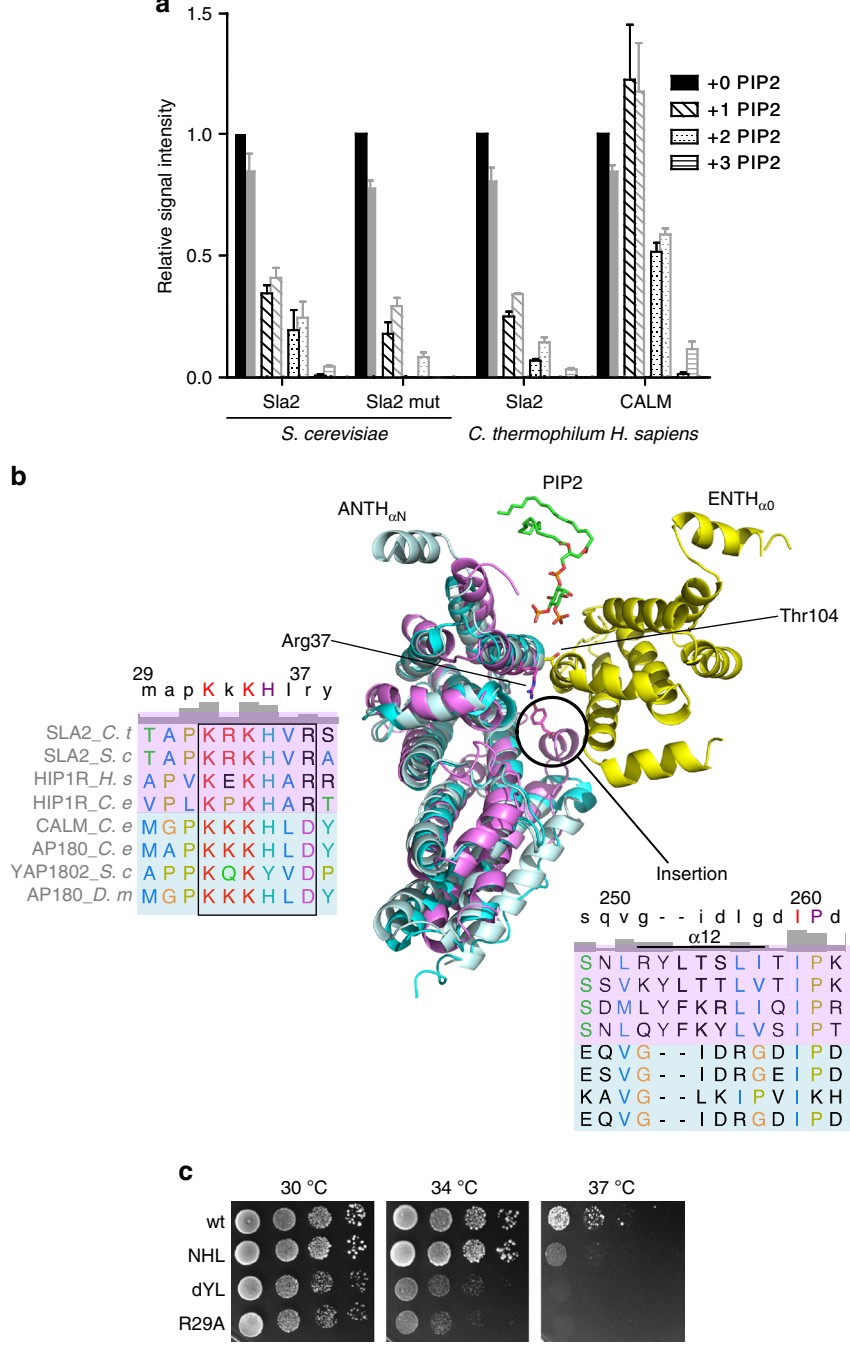

**Fig. 3** Evolutionarily conserved features in Hip1R subfamily of ANTH domains. **a** Native MS was used to investigate PIP2 binding to several members of the ANTH family: $ANTH_{Sla2}$, $ANTH_{Sla2,mut}$ (four residues to Ala in the canonical PIP2-binding site, see Constructs) and CALM from indicated species. Signal intensities from mass spectra of at least three independent measurements were summed over all charge states, normalized to the corrected signal of the unbound protein and the average of the relative signal intensities and their standard deviations were plotted (back). After correction for unspecific clustering, relative signals of unbound and PIP2-bound ANTH domains are obtained (front). Signal for proteins with three attached PIP2 molecules observed in raw spectra disappear after correction. **b** Composite model of the ENTH and $ANTH_{Sla2}$ complex based on the low-resolution EM structure[14] and the high-resolution X-ray crystal structures presented here (ENTH2/PIP2, yellow, and superimposed ANTH domains of Sla2 (violet), AP180 (cyan), and CALM (pale cyan). Several elements contributing to the interface are highlighted, including the inserted α helix in the $ANTH_{Sla2}$ domain, the proximate Arg 37 of $ANTH_{Sla2}$ and Thr 104 from ENTH, and the location of PIP2 from the secondary site of the ENTH2/PIP2 crystal structure. **c** Growth defects of Sla2(1–360) strains mutated in Sla2/Hip1R ANTH-specific features. Tenfold serial dilutions of sla2Δ strains expressing Sla2(1–360) wt, Sla2(1–360) NHL with α8–α9 loop replaced by residues NHL as occurring in Yap1802, Sla2(1–360) dYL deletion of conserved Tyr 252 and Leu 253 and Sla2(1–360) R29A were incubated on SC-Ura plates at 30, 34, and 37 °C for 2 days

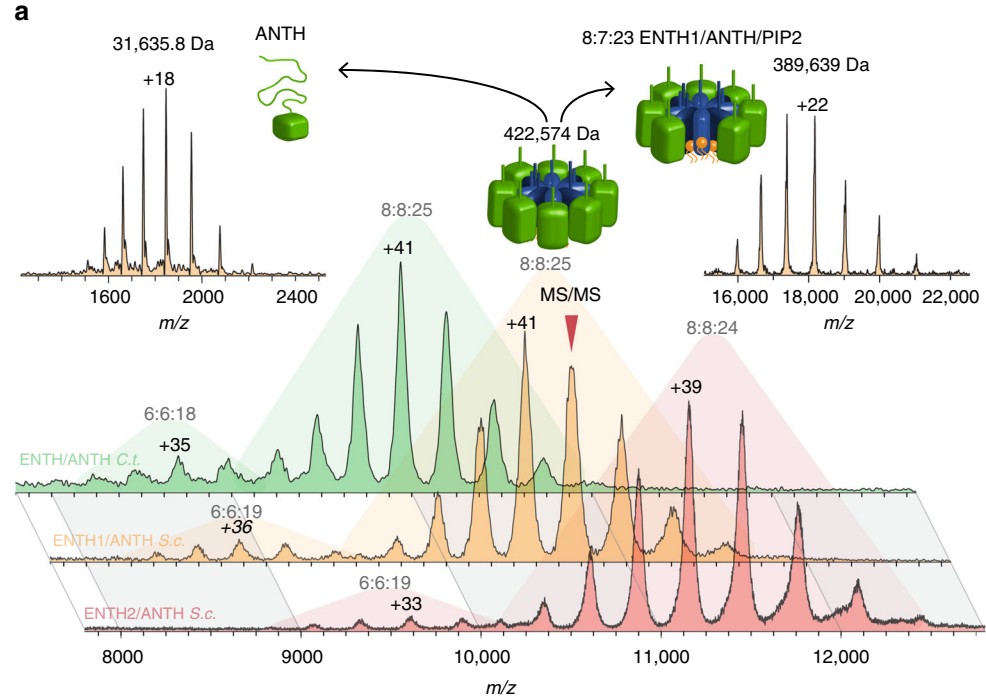

**Fig. 4** ENTH/ANTH/PIP2 complex formation in fungi 6:6:~18 and 8:8:~24 ENTH/ANTH/PIP2 complexes are the observed stoichiometries in native MS measurements. A cartoon of the most prominent complex is shown with ANTH in green and ENTH in blue. Complexes from *C. thermophilum* (green) and *S. cerevisiae*, containing ENTH1 (orange) or ENTH2 (red), show the same stoichiometries, signal ratios and dissociation pathways in collision-induced dissociation (CID) MS/MS. Here, the dissociation of the +40 charged 8:8:25 ± 3 ENTH1/ANTH$_{Sla2}$/PIP2 complex into partially unfolded ANTH$_{Sla2}$ (top left, cartoon shows green ANTH domain) and a residual 8:7:23 ± 1 complex (top right, showing also a cartoon of the remaining complex) is depicted. The annotation shows the stoichiometry (ENTH:ANTH:average PIP2 number), charge state of the main peaks, and average experimental masses. Ranges of PIP2 numbers, statistical errors, and an average FWHM value rating the MS resolution of all complexes can be found in Supplementary Table 2. ANTH dissociation in CID MS/MS measurements of 6:6:18 ± 1 ENTH2/ANTH$_{Sla2}$/PIP2 complexes from *S. cerevisiae* is presented in Supplementary Fig. 4b

could be conserved in mammals[13]. To verify evolutionary conservation of ENTH/ANTH/PIP2 complexes, a mixture of human epsin-1 ENTH domain with human Hip1R ANTH domain and PIP2 was analyzed by native MS, showing less homogeneity with a larger variety of oligomers (Fig. 5b). However, the hetero 16-mers observed with *S. cerevisiae* domains were also present in samples with human domains. The majority of human ENTH/ANTH complexes consist of a combination of six epsin ENTH molecules and a range of ANTH Hip1R molecules. Contrary to the fungal samples, the ENTH hexamer is already observed in the presence of PIP2 alone (Fig. 5c). A minimum of six PIP2 molecules is required for human ENTH hexamer formation. The appearance of the hexameric ENTH core in the human homolog suggests higher stability compared to fungal homologs.

To confirm existence of the hexameric human ENTH core with PIP2 in solution and to gain further insight into its structure, small-angle X-ray scattering (SAXS) was performed on human ENTH (Supplementary Table 3). In the presence of PIP2, a uniform hexameric species is observed (Fig. 6a and Supplementary Table 3). Using the ENTH2/PIP2 dimeric interface from the *S. cerevisiae* ENTH2/PIP2 crystal structure as a rigid body, and representing the C-terminal parts missing in the crystal structure as dummy residue chains, a model for the hexamer was constructed yielding an excellent fit to the experimental data (Fig. 6b and Supplementary Fig. 4d). Three ENTH domains are oriented with α0 helices in plane of the cell membrane, in accordance with previous observations[5,8,25]. For the three other ENTH domains, the PIP2 molecule is still oriented in plane with the cell membrane (Fig. 6c), but α0 is pointing away, available to interact with additional ENTH domains to form larger oligomers. Similarly, the interface involving Thr 104 is alternatively used for

hexamer formation as well as further oligomerization through phospholipid interfaces.

The potential secondary PIP2-binding sites are also available for docking of ENTH or ANTH domains bringing in their own PIP2 molecule to make a larger assembly.

**Characterization of phospholipid-dependent complex formation.** We used thermal denaturation aggregation assays by dynamic light scattering (DLS), to investigate stability differences between fungal and human ENTH/ANTH/PIP2 complexes. ENTH1 from *S. cerevisiae* in the presence of PIP2 shows a relatively low mid-aggregation temperature ($T_{agg}$) of 33 °C (Fig. 6d), which increases to 37 °C when ANTH$_{Sla2}$ is added. In contrast, the hexameric human ENTH core shows a $T_{agg}$ of 45 °C in the presence of PIP2, which is more stable than the assembled fungal ENTH/ANTH complex. The optimal body temperature for *H. sapiens* is 37 °C, and the thermostability assay indicates that human ENTH domain alone will form a stable complex at that temperature, in contrast to ENTH domains from *S. cerevisiae*.

To test whether PIP2 binding is the driving force for complex formation, rather than direct protein–protein interactions, we looked for cross-species complexes from *C. thermophilum* and *S. cerevisiae* by native MS, ITC, and DLS (Supplementary Fig. 5). Indeed, such complexes can be formed and are surprisingly robust as measured by ITC ($K_D$ = 450 nM ± 59 nM) (Supplementary Fig. 5c, d), despite low sequence identity. We also investigated whether the ANTH$_{Sla2}$ domains from *S. cerevisiae* and *C. thermophilum* bind to the human epsin ENTH core. The human-fungi cross-species assemblies show degenerated formation of PIP2-dependent complexes, with the 6:6 complex as the

largest assembly observed by native MS (Supplementary Fig. 5b). These cross-species measurements confirm that human ENTH forms a stable hexameric core in the presence of PIP2, which can be decorated by ANTH domains through PIP2 interfaces. However, the fungal ANTH cannot open up the hexameric core to create larger assemblies as in the human ENTH/ANTH system.

## Discussion

Clathrin-mediated endocytosis involves a diverse set of adapter proteins that help to curve the cell membrane and to recruit clathrin and cargo. The initial stages of vesicle coat assembly involve many adapter proteins and may involve many alternative assemblies[26]. Epsin and the phospholipid PIP2 are essential for clathrin-mediated endocytosis[22,27], and we show how assemblies

are formed between the ENTH domain of epsin and the ANTH domain of Sla2 in fungi. The ENTH and ANTH domains both contain two PIP2-binding sites, as determined by native MS. It has to be noted that the mature ENTH/ANTH 8:8 complexes contain only $24 \pm 3$ PIP2 molecules, which is much less than 32 PIP2 expected based on the number of available PIP2-binding sites. This indicates that PIP2 is shared between domains, and acts as a glue to bring these adapter proteins together. Recently, a similar gluing mechanism has been discovered using native MS for specific lipids that strengthen the oligomerization of membrane proteins[28]. This gluing mechanism is illustrated further by the X-ray crystal structure of the ENTH2 domain from *S. cerevisiae* in complex with PIP2, where one PIP2 molecule is sandwiched between two ENTH domains. The two ENTH2 domains glued together are structurally distinct. The ENTH2 domain that captured the PIP2 molecule in the canonical-binding pocket has an N-terminal region that is folded into an α helix. In contrast, the ENTH2 domain that associates to the other side of the PIP2 molecule through an interface that includes residue Thr 104 has an unfolded N-terminus. This ENTH2 domain could capture another PIP2 molecule on the cell membrane, which in turn would fold the N-terminal region into an α helix. This structure therefore not only confirms the presence of an allosteric switch upon PIP2 binding, but also shows how this binding event can trigger further oligomerization of ENTH domains at the cell surface. It is intriguing that this oligomerization process could include both ENTH1 and ENTH2 domains, and that the ANTH$_{Sla2}$ domain from *S. cerevisiae* does not show preference for either ENTH domain. This indicates that both epsins Ent1 and Ent2 can contribute to the assembly of the epsin/Sla2 complex, and qualifies the redundancy that is observed between epsin family members.

So far, it seems the formation of the PIP2-driven complex between ENTH and ANTH domains is limited to Sla2/Hip1R ANTH domains[12,13,21]. Attempts to form PIP2-driven assemblies between ENTH domains and AP180/CALM subfamily members by native MS were unsuccessful. A crystal structure of the Sla2 ANTH domain from *C. thermophilum* reveals that Hip1R-related ANTH domains have two evolutionarily conserved insertions that contribute to the epsin/Hip1R interface to enable PIP2-dependent ENTH/ANTH$_{Sla2}$ complex formation. Especially the insertion of an α helix around Tyr 252 provides surface complementarity in the ENTH/ANTH$_{Sla2}$-binding interface, compared to the structures of the ANTH domains of AP180/CALM proteins. Superimposition of the ENTH domain with the secondary PIP2-binding site shows how the PIP2 molecule fits into the ENTH/ANTH$_{Sla2}$ interface close to the PIP2-binding pocket on the ANTH$_{Sla2}$ molecule (Fig. 3b). The residues Thr 104 from ENTH and Arg 37 from *C. thermophilum* ANTH$_{Sla2}$ are facing each

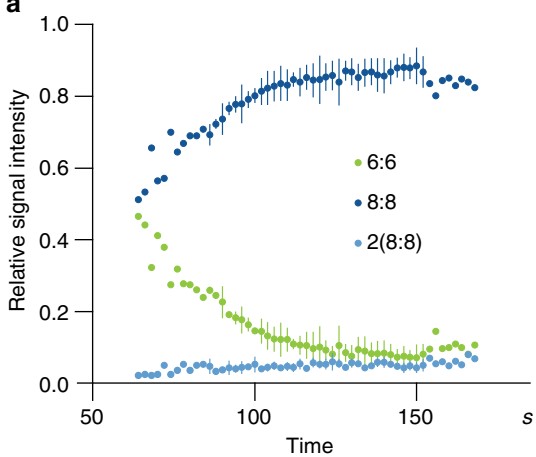

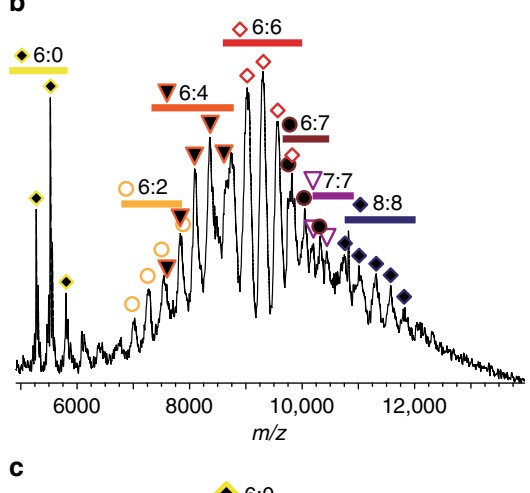

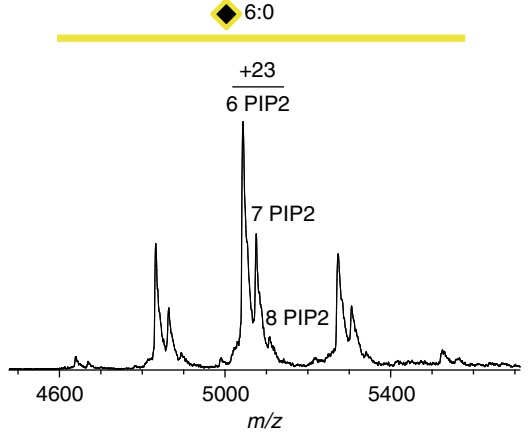

**Fig. 5** Dynamics of ENTH/ANTH/PIP2 complex formation in fungi and human assemblies. **a** Time course of ENTH2/ANTH$_{Sla2}$/PIP2 (*S. cerevisiae*) complex formation. Components were mixed, injected into the electrospray capillary and the spectra monitored over time. Relative signal intensities for 6:6 (green), 8:8 (dark blue), and dimers of 8:8 complexes (light blue) were determined and plotted against time ($n = 3$). The signal of the 6:6 complex drops within 2 min after mixing the complex components, while the signal of the 8:8 complex increases, suggesting a transition between these forms. The signal of the 8:8 dimer remains constant, ruling out aggregation effects. Average data of three independent measurements, error bars (standard deviation) are shown for data points with N = 3 **b** Native MS shows oligomerization of human epsin-1 ENTH and Hip1R ANTH in various stoichiometries, ranging from 6:0 to 8:8 with 6:6 being the main species. **c** Human epsin-1 ENTH domain forms hexamers (6:0) with at least six PIP2 molecules also in absence of Hip1R

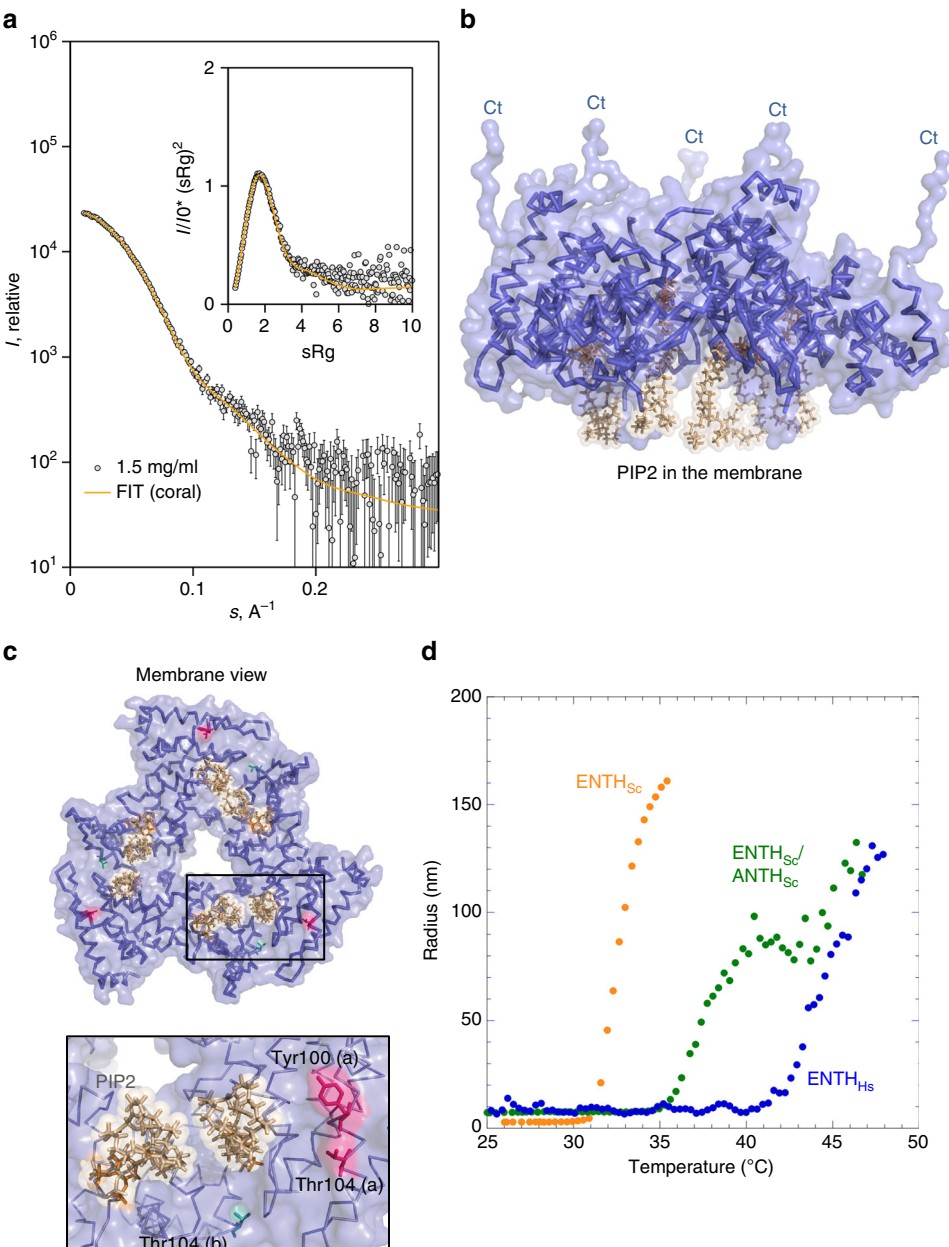

**Fig. 6** Human ENTH forms a thermally stable hexamer with a predicted membrane-binding interface. **a** SAXS modeling of human ENTH in the presence of PIP2. SAXS data recorded for human ENTH in solution are shown (gray circles) along with a fit to the rigid-body model refined against the SAXS data with P3 symmetry using CORAL (orange solid line). Experimental errors are from counting statistics on the Pilatus 2M detector and propagated through the data reduction process as standard errors in the scattering intensities. The $\chi^2$ for the fit is 1.05. The inset shows the dimensionless Kratky plot representation of the SAXS data and the same fit. **b** Backbone and surface representation of the SAXS refined rigid body model. C-terminal (Ct) residues not observed in the crystal structure of the tandem domains are modeled as dummy residues by CORAL. P3 symmetry was enforced and the tandem ENTH domains with bound PIP2 used as rigid bodies. **c** 90° rotation of the model, demonstrating that the bound PIP2 molecules are all located on one side of the protein, consistent with a membrane-binding interface. The Thr 104 residues involved in ENTH/ENTH homodimerization are colored pink, whereas the Thr 104 residues present on the surface of the hexamer are colored green. **d** Hydrodynamic radius as a function of temperature measured by DLS (ENTH, *S. cerevisiae*, orange; ENTH/ANTH complex, *S. cerevisiae*, green; hexameric ENTH core, *H. sapiens*, blue)

other and have been shown by mutagenesis to be essential for the formation of the ENTH/ANTH$_{Sla2}$ complex[14]. In addition, residue Arg 37 can toggle its side chain between the canonical PIP2-binding site of ANTH domains, and the ENTH/ANTH$_{Sla2}$ interface.

The biophysical experiments presented here have revealed that there is a difference in epsin and Sla2/Hip1R ENTH/ANTH complex assembly between fungal and human proteins. There is a general sequence of events revealed by native MS (Fig. 7). After

PIP2 binding, ENTH domains form a core that is decorated on the outside by ANTH domains. Many different assemblies may occur, but in fungi the ENTH core alone is not stable enough to be maintained. The ENTH and ANTH domains cluster to yield a homogeneous 8:8 complex (Fig. 7a). In humans, the ENTH hexamer core is stable, but once it is decorated with ANTH, many different clusters may form (Fig. 7b). Our in vitro data are in nice agreement with cell biological studies on epsins and Sla2/Hip1R proteins in yeast[12], amoebae[21,23], and mammals[13]. In *S.*

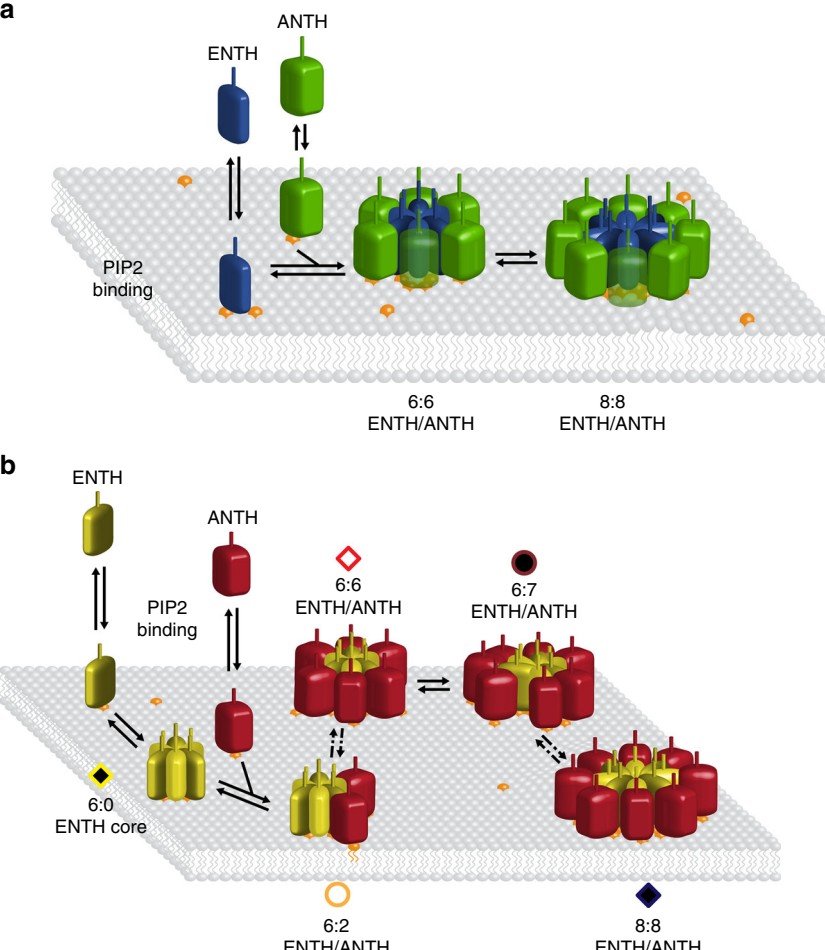

**Fig. 7** Schematic models of PIP2 binding initiating clustering of ENTH and ANTH domains. **a** In *S. cerevisiae* and *C. thermophilum* ENTH (blue) and ANTH (green) domains bind PIP2 (orange) in the membrane and cluster to hetero 12-mers with the stoichiometry 6:6 that consequently can be transformed to more stable hetero 16-mers with the stoichiometry 8:8. **b** Human ENTH domains (yellow) bind PIP2 (orange) and cluster to homo 6-mers. ANTH domains (red) bind in different stoichiometries. 6:6 hetero 12-mers are the most abundant species, but a transition to larger complexes, up to hetero 16-mers can be observed. Symbols assigning the different complex stoichiometries are chosen as in the mass spectrum of Fig. 5b. Dashed arrows indicate that not all complex stoichiometries are represented in this model

*cerevisiae*, Sla2 ANTH is needed for the stable binding of epsin ENTH to the endocytic site and both proteins are essential for actin-dependent endocytosis[12,14]. In contrast, mammalian epsin ENTH binds to the endocytic site in the absence of Hip1R[13] and is necessary for Hip1R recruitment to the endocytic site. Consequently, an absence of epsin has more impact on endocytosis than the lack of Hip1R, suggesting epsin ENTH is crucial in PIP2-driven adapter protein complex formation in mammalian endocytosis. The independent formation of higher oligomers of the human ENTH domain alone was also observed in giant unilamellar vesicles by fluorescence microscopy[25].

Autonomous assembly of human ENTH domains driven by PIP2 creates a scaffold that could contribute to membrane bending and scission[29]. Since the interactions between the ENTH domains are dominated by PIP2 interfaces, different epsins can contribute to the scaffolding, explaining why only a complete knockout of all epsins shows a phenotype that disrupts CME[13]. We propose that mammalian ENTH domains not only curve the membrane by the insertion of an amphipathic helix, but also can form a membrane scaffold when PIP2 is concentrated at an endocytic site. This proposal would also give further credence to molecular crowding mechanisms that are aided by the accumulation of phospholipid-dependent assemblies of adapter

proteins[30]. Further studies on formation of complexes between PIP2-binding adapter proteins will hopefully provide insights into seeding and maturation processes that help to dedicate vesicles to certain endocytic or exocytic pathways.

## Methods

**Chemicals**. Phosphatidylinositol-4,5-biphosphate diC8 (PIP2) was purchased from Echelon. n-dodecyl β-ᴅ-maltoside (DDM) was purchased from Anatrace.

**Constructs and plasmids**. The coding sequence for the ANTH and ENTH homologs from *Chaetomium thermophilum* were amplified from an RNA library kindly provided by Peer Bork's laboratory (see constructs in Supplementary Table 4); EMBL Heidelberg[31] using the QIAGEN OneStep RT-PCR Kit. The reverse transcription was done at 50 °C for 30 min and the second round PCR was performed in the presence of solution Q and a touchdown procedure was introduced starting at 72 °C (see primers in Supplementary Table 5). The DNA coding sequence was cloned in pETM-11/LIC vector (EMBL).

Human CALM coding sequence has been purchased from Addgene (Plasmid #27691) and the ANTH domain was cloned in pETM-11/LIC vector for expression with an N-terminal 6xHis-tag. Sequence coding for ANTH domain of human Hip1R (aa 1–300) was PCR amplified from human cDNA and cloned into the *NcoI-XhoI* sites of pETM30 (EMBL). Sequence coding for ENTH domain of human EPN1 (aa 1–158) was PCR amplified from human cDNA (with the additional codon for glycine after START codon) and cloned into the *BamHI-XhoI* sites of pETM30. Sequence coding for ANTH domain of Yap1802 (aa 1–272) was PCR amplified from yeast gDNA and cloned into the *NcoI-XhoI* sites of pETM30.

**Protein production and purification**. Recombinant human Hip1R ANTH, human ENTH and the yeast Yap1802 ANTH domains were expressed in *E. coli* BL21 DE3 (Novagen) as GST fusion proteins containing an N-terminal His-tag followed by a TEV (Hisx6-TEV) cleavage site. Recombinant human ANTH domain from CALM and *C. thermophilum* ANTH and ENTH domains were expressed in *E. coli* BL21 DE3 pLYS (Novagen) as N-terminal Hisx6-TEV-tagged proteins. 4 L flasks containing 800 ml cultures in LB media were grown at 37 °C until an optical density (OD = 600) of 1.0. After induction with 0.5 mM IPTG, the cultures were moved to an incubator at 20 °C and harvested after 4 h. Protein expressing cells were harvested by centrifugation (3500 x *g* for 30 min at 4 °C). The cell pellet was lysed by sonication in the presence of 1 mg/ml DNase (400 K units) in 10 mM Tris-HCl pH 7.5. Lysed cell extract was centrifuged (45,000 x *g*, 45 min at 4 °C) and supernatant containing His-tagged proteins were purified by nickel-nitrilotriacetic acid (Ni-NTA) purification (Qiagen). Protein was eluted in a final elution buffer of 20 mM Tris pH 8.0, 300 mM NaCl, 250 mM imidazole. Excess of TEV protease was added to the imidazole eluted fractions for cleavage of the Hisx6-GST and Hisx6 tags (1 mg/ml TEV per liter of culture). Digestion was performed by dialysis at 4 °C overnight against 5 L of 20 mM Tris pH 8.0, 250 mM NaCl and 1 mM DTT. To remove the tags, the dialyzed fractions were subjected to a second Ni-NTA and the flow-through was concentrated to 5 mg/ml to be injected in a size exclusion chromatography (SEC). SEC was performed using an Äkta liquid chromatography system (Amersham Biosciences) and S75 10/300 GL (Tricorn) column (GE Healthcare) in 20 mM Tris-HCl pH 8.0 and 250 mM NaCl.

**Isothermal titration calorimetry**. ITC was performed at 25 °C using MicroCal VP-ITC calorimeter (GE Healthcare). An aliquot of 100 μM ENTH was titrated into 10 μM ANTH contained in the cell. ITC buffers: 20 mM Tris-HCl pH 8.0 (for *S. cerevisiae* ANTH) or 20 mM sodium phosphate pH 8.0 (for ANTH$_{Ch}$) with 250 mM NaCl supplemented with 200 μM PIP2.

**Dynamic light scattering**. Measurements have been performed using DynaPro Nanostar (serial no. 323-DPN, Wyatt Technology Corporation). Data have been processed using Dynamics v.7 software. 30 μM protein and PIP2 were mixed and incubated overnight at 4 °C. Samples were filtered through 0.22 μm centrifugal filters (Millipore). The acquisition time was 3 s with a total of 20 acquisitions.

**Crystal structure determination**. The ENTH2 domain from *S. cerevisiae* in complex with PIP2 was prepared with a protein concentration of 6 mg/ml pre-mixed with 200 μM PIP2 and incubated at 4 °C for 2 h. The sample was filtered through 0.22 μm centrifugal filters (Millipore). Protein crystals were obtained by vapor diffusion in a hanging drop setup using Limbro Plates (Hampton Research). For the crystallization drop, 1 μl of the complex was mixed with 1 μl of a mother liquor containing 0.1 M MES, pH 6.5, 0.1 M NaCl, 1.45 M ammonium sulfate. To obtain a complete data set to 3.35 Å resolution, X-ray data from three cryo-cooled crystals soaked in paraffin oil were combined (see Table 1). The ENTH2 domain from *S. cerevisiae* was crystallized with a protein concentration of 8 mg/ml in a buffer containing 10% (v/v) isopropanol, 20% PEG 4000 and 0.1 M Hepes pH 7.5. The ENTH1 domain from *S. cerevisiae* was crystallized with a protein concentration of 11 mg/ml in a buffer containing 0.2 M CaCl$_2$, 25% v/v MPD and 0.1 M Tris pH 8.5.

The ENTH1 and ENTH2 domain structures were solved by molecular replacement using PHASER[32], with search models of the corresponding domains present in the PDB (*Rattus norvegicus* ENTH code 1EDU and *S. cerevisiae* ENTH2 code 4GZC). The structure was refined with REFMAC5[33] and manually rebuilt with Coot[34], and the statistics are reported in Table 1. The ENTH2/PIP2 structure was solved by molecular replacement using the high-resolution structure of ENTH2 determined at 1.8 Å resolution (Table 1). The structure was refined with Phenix[35] and manually rebuilt with Coot, resulting in a final $R_{factor}$ of 27.9% ($R_{free}$ = 30.3%). The stereochemistry was checked with Molprobity[36], indicating good overall geometry with only 1.8% of the residues in disallowed regions of the Ramachandran plot. Structure diagrams were prepared with Pymol (The PyMOL Molecular Graphics System, Version 1.7.x, Schrödinger, LLC) and Chimera[37].

The Sla2 ANTH domain of *C. thermophilum* (ANTH$_{Ch}$) was crystallized with 0.2 M potassium bromide, 0.1 M Tris pH 7.5, 8% (w/v) PEG 20 K, 8% (w/v) PEG 550 MME. A single, cryo-cooled crystal of ANTH$_{Ch}$ diffracted to 1.8 Å at the MASSIF beamline[38] and belonged to space group *P*2$_1$ (Table 1). The structure was solved by molecular replacement using a structure prediction from the BAKER-ROSETTA SERVER through the Protein Structure Prediction Center CASP11[39] as a search model. A total of 150 in silico designed models that were based on the amino acid sequence of Hip1R were provided by CASP11. One model designed by the Baker-Rosetta team gave a marginal solution, placing two molecules in the asymmetric unit of the *P*2$_1$ crystal form. The CASP model was derived from the CALM crystal structure (PDBID 3ZYM), but it had an RMSD of 2.0 Å for 219 residues when superimposed on the original model. Molecular replacement and model building were performed using PHASER[32] and ARPWARP[40]. Subsequent manual inspection using Coot[34] and refinement with REFMAC5[33] resulted in a refined structure with an $R_{factor}$ of 16.7% ($R_{free}$ = 22%). The stereochemistry was checked with Molprobity[36], indicating good overall geometry with only 0.5% of the residues in disallowed regions of the Ramachandran plot.

**SAXS**. Human ENTH hexamer was assembled at 80 μM protein concentration in the presence of 20 mM Tris-HCl pH 8.0, 250 mM NaCl and 0.2 mM PIP2 and incubated overnight at 4 °C. As a control, 80 μM human ENTH with no PIP2 added was used. Synchrotron radiation X-ray scattering data were collected (EMBL P12, PETRA III, DESY, Germany)[41] (Supplementary Table 3) with a PILATUS 2 M pixel detector (DECTRIS, Switzerland) (20 × 0.05 s frames). Solutions of human ENTH (0.4–1.5 mg/ml) were measured through a capillary (20 °C). The sample-to-detector distance was 3.1 m, covering a range of momentum transfer 0.01 ≤ s ≥ 0.46 Å$^{-1}$ (s = 4π sin$\theta/\lambda$). Frame comparison showed no detectable radiation damage. Data from the detector were normalized, averaged, buffer subtracted, and placed on an absolute scale that is relative to water, according to standard procedures. All data manipulations were performed using PRIMUSqt and the ATSAS software package[42]. The forward scattering $I(0)$ and radius of gyration, $Rg$ were determined from Guinier analysis: $I(s) = I(0)\exp(-(sRg)2/3))$. The indirect Fourier transform method was applied using the program GNOM[43] to obtain the distance distribution function $p(r)$ and the maximum particle dimensions $D_{max}$. Molecular masses (MMs) of solutes were estimated from SAXS data by comparing the extrapolated forward scattering with that of a reference solution of glucose isomerase (Hampton), the hydrated particle/Porod volume $V_p$, where molecular mass is estimated as 0.588 times $V_p$, and from the excluded solvent volumes, $V_{ex}$ is obtained from DAMMIF[44] through ab initio modeling. Rigid body modeling was performed using the program CORAL[42].

**Native mass spectrometry**. Prior to native MS analysis[45], purified proteins were buffer exchanged to 300 mM ammonium acetate (PN 431311, 99.99% purity, Sigma-Aldrich) and 1 mM DL-dithiothreitol (PN 43815, 99.5% purity, Sigma-Aldrich), pH 8.0, via centrifugal filter units (Vivaspin 500, MWCO 5000 and 10000, Sartorius) at 13,000 x *g* and 4 °C. Complexes were assembled after buffer exchange by mixing 10 μM ENTH, 10 μM ANTH and 60 μM PIP2 (Phosphatidylinositol-4,5-biphosphate diC8, Echelon). Thus, soluble PIP2 was used at a submicellar concentration that is in the range of its physiological concentration in the cell[46].

ESI capillaries were produced in house. Therefore borosilicate capillaries (1.2 OD, 0.68 ID, World Precision Instruments) were processed in a two-step program with a micropipette puller (P-1000, Sutter instruments) using a squared box filament (2.5 × 2.5 mm, Sutter instruments) and subsequently gold-coated using a sputter coater (Q150R, 40 mA, 200 s, tooling factor of 2.3 and end bleed vacuum of 8 × 10$^{-2}$ mbar argon, Quorum Technologies).

Native MS on protein complexes was performed on a QToF 2 (Waters and MS Vision) modified for high mass experiments[47] with nanoESI in positive ion mode. The gas pressures were 10 mbar in the source region and 1.1 × 10$^{-2}$ mbar xenon (purity 5.0) in the collision cell[48,49]. Mass spectra were recorded with applied voltages for capillary, cone, and collision of 1.35 kV, 120–150 V, and 30–60 V, respectively, optimized for good resolution and minimal complex dissociation. Complexes were analyzed in MS/MS by ramping collision voltages from 10 to 400 V in order to eject protein subunits. Raw data was calibrated with 25 mg/ml cesium iodide spectra of the same day with the software MassLynx (Waters). MassLynx and Massign[50] were used to assign peak series to protein species.

Lipid binding was studied on an LCT ESI-ToF system (Waters and MS Vision)[47] with direct infusion using the gentlest ionization possible (capillary 1.4 kV, cone 100–120 V, extraction cone 0 V, 6.5 mbar source pressure). Samples were prepared by adding 60 μM PIP2 (Phosphatidylinositol-4,5-biphosphate diC8, Echelon) to the protein of interest (10 μM). For ENTH domains, cytochrome *c* from equine heart (PN 129021, Sigma-Aldrich) and for ANTH domains, carbonic anhydrase isoenzyme II from bovine erythrocytes (PN C2522, Sigma-Aldrich) was used as reference protein to test for unspecific clustering as described[18]. If there was signal overlap of the ENTH or ANTH domain with signal of the PIP2-bound reference protein, this charge state was excluded from the calculation of the ratio of unspecific binding. As an approximation the unspecific binding that was determined using the remaining charge states of the reference protein was subtracted from overlapping signals and the residual signals of the overlapping peaks were considered for further analysis. Relative peak intensities were used to determine the ratio of lipid-bound control protein to non-bound control protein. This ratio was used to correct peak intensities and visualize the data using GraphPad Prism (GraphPad Software). Cooperativity of the two binding sites was assessed by reviewing the mathematical relation of microscopic (binding site) and macroscopic (apparent) dissociation constants of independent binding sites (see Fig. 1c). As shown in[20] these are related by:

$$K_{D,1} = \frac{k_{d,1} \times k_{d,2}}{k_{d,1} + k_{d,2}} \qquad (1)$$

$$K_{D,2} = k_{d,1} + k_{d,2} \qquad (2)$$

A lack of microscopic constants $k_{d,1}$ and $k_{d,2}$ fulfilling the required condition for macroscopic constants is an indication for dependence of PIP2-binding sites. Macroscopic dissociation constants in the same order of magnitude for both binding events suggest positive cooperativity.

All errors given for native MS data refer to the standard deviation and were based on at least three independent measurements.

**Yeast strain and plasmids**. Yeast strain MKY0764 (MATa, his3Δ200, leu2-3,112, ura3-52, lys2-801, sla2:natNT2) was transformed by pRS416-based centromeric plasmid[12] expressing indicated variants of Sla2 aa 1–360 fragments[51] (generated by overlapping PCR with mutagenic primers). The expression levels of protein constructs were assessed by immunoblotting with anti-HA antibodies (Covance MMS-101R) at 1:1000 dilution. The secondary antibody used was anti-mouse HRP conjugate (BIO-RAD, 1706516) at 1:3000 dilution.

**Data availability**. The atomic coordinates for the ENTH2/PIPI2 complex (PDBID 5ON7), the ENTH1 structure (PDBID 5ONF), and the ENTH2 structure (PDBID 6ENR) from the *S. cerevisiae* as well as the Sla2 structure from *C. thermophilum* (PDBID 5OO7) are deposited at the Protein Data Bank. Other data are available from the corresponding authors upon reasonable request.

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

## Acknowledgements

We thank Rob Smock, Daniel Lietha, and Michel Koch for useful comments on the manuscript. We acknowledge the staff of the EMBL beamlines P12 and P14 as well as the MASSIF beamline at the ESRF, and the sample preparation and characterization (SPC) facility of EMBL at PETRA III (DESY, Hamburg) for assistance. M.G.-A. and A.G. were supported by the EMBL Interdisciplinary Postdoc Program (EIPOD) under Marie Curie COFUND actions. The Heinrich Pette Institute, Leibniz Institute for Experimental Virology is supported by the Free and Hanseatic City of Hamburg and the Federal Ministry of Health. J.H. and C.U. are funded by the Leibniz Association through SAW-2014-HPI-4 grant. M.K. is funded by the Swiss National Science Foundation (31003A_163267).

## Author contributions

R.M. conceived the work together with C.U. and M.K. M.G.-A. produced proteins and samples for crystallization, SAXS, and native mass spectrometry, determined the Sla2 ANTH crystal structure, designed and performed biophysical experiments, and interpreted data. J.H. performed mass spectrometry experiments and interpreted the data together with C.U. M.S. performed growth experiments in *S. cerevisiae*. A.G. produced proteins for native mass spectrometry and the ENTH1 and ENTH2/PIP2 crystals. H.D.T. M. performed SAXS analysis and interpreted the data together with D.I.S. R.M. wrote the manuscript with input from all authors.

## Additional information

**Competing interests:** The authors declare no competing financial interests.

