## [Peer Review File · Nature Communications]

Reviewers' comments:

Reviewer #1 (Remarks to the Author):

This manuscript by Garcia-Alai is entitled Epsin and Hip1R form assemblies through phospholipid interfaces and yet the majority of the data, certainly the structural data rather than modelling concerns not the mammalian proteins as the title suggests but their fungal equivalents. This is inappropriate and seems to be an increasing trend on the yeast field to refer to mammalian proteins in the title instead of the fungal proteins on which the work is actually mainly done. The yeast data is good enough to stand on its own and the mammalian work is the weakest part of the paper as it is based largely on modelling studies and not compelling data. I think the title should be changed to reflect this. I think there would be a good case for removing the flakey sections on the mammalian proteins towards the end of the Results section but if the authors wish to retain it then they should back it up with further data. regardless of which route is chosen, more information and explanation and possibly further clarifying data is needed on the status of PIP2 native mass spec part

In general however, the paper is well written, the data presented well and the crystallography done properly and the native mass spec, which is a cornerstone of the work, is relatively new and interesting. There is a tendency to not discuss the mammalian literature sufficiently/selectively when it is referred to.

I have a serious technical issue concerning the use of PIP2

I also have a number of requests and questions that would need answering in a revised manuscript

My major technical concern focuses on the use of PIP2 in native mass spec. I can find no information as to whether it is in liposomes and indeed whether it is neat PIP2 or PIP2 mixed with the normal physiological other lipids. Indeed I am not even sure if it is PIP2 with long medium short or no (ie headgroup) acyl chains. The concentrations mentioned seem very large to me hence many sites would be saturated , which would not be in vivo. In the cell estimates of PIP2 concentration at plasma membrane are 1-5% but this does not take into account all that is bound by other proteins of both trafficking and especially signalling backgrounds. At the high concentrations used, does the PIP2 form aggregates, liposomes or membrane-like fragments. This needs to be characterised and I would suggest the experiments performed with relevant lipid mixtures

Throughout such as on page 4, but also in many other places, the authors do not state whether they are dealing with the fungal or human proteins. Many readers will be unsure of the origins of the proteins. This is important since there is a fundamental difference in endocytosis in yeast and mammals concerning many aspects including the need for actin and AP2 (the central controller in mammals not needed in yeast). Please make it clear by stating each time if it is a yeast or a human protein.

P5&6 One decimal place for resolution is appropriate. There should be electron density shown for each structure of a sample piece and of key functional regions and ligands so that readers can judge whether or not the authors interpretations are appropriate. At 3.4Å a lot of things are unclear and often ambiguous. It is not acceptable to not show the electron density. A PDB check file could also be included.

P7 Why do the authors insist on calling CALM PICALM? PICALM is an invention of the alzheimers disease community – we all know it bind PhosphoInositides. CALM has been published by several groups to have a 1-20uM affinity for PIP2 in membranes using a wide variety of well tried and

tested techniques – what do the modelling studies show here? Is Mass spec a more reliable way of determining K_Ds? The conditions are certainly not physiological and it is well known that going from 100 to 250mM NaCl can move the K_D of epsin for PIP₂ from 1μM to unmeasurably weak. Here 300mM NH₄OAc is used and this could greatly affect affinity measurements. What are the apparent K_Ds nearer to physiological i.e. 160mM NH₄OAc

P8 In order to confirm whether an alpha helix exists in HIPR-like ANTH domains, it would be necessary to carry out CD ± liposomes and I can see no indication this was done. Also does the Nterminus predict as an amphipathic helix – use Bruno Antonny's server (heliquest).

P9, removal of an ANTH occurs due to collision in MS as I understand it. Do more come off with more time? How relevant do the authors consider this and why since collisions don't occur in the cell.

P10 Why is full length and not ANTH HIPR used?

P10 The mammalian protein oligomers are extremely heterogeneous – why should the 6 ENTH based complexes be considered to be biologically relevant? What fraction actually has a hexamer ENTH at its core?

In general the data on mammalian proteins seems rather preliminary and needs firming up.

In mammalian cells, as opposed to fungal cells, the roles of epsin and HIPR are proposed to be very different. Hip1R is meant to interact with clathrin light chains distant from the membrane surface and epsin has been extensively referenced as causing membrane curvature and more recently, scission, as deletion of all epsins results in the production of bunches of grapes (McMahon lab). How does the formation of multimers of some size in mammals fit with these models. The authors should explain these points and how their speculative model fits into the wealth of existing data.

Reviewer #2 (Remarks to the Author):

The authors previously reported that yeast clathrin adaptor proteins, Sla2 and Ent1, co-assemble in a PIP₂-dependent manner (Skruzny 2015). In this paper, the authors report biophysical analysis of this interaction. Although extensively studied, there are many concerns to be addressed.

1) The authors claim that the ENTH and ANTH domains both bind two PIP₂ by native MS analysis (Figures. 1 and 3). It is desirable to show ITC data to support the native MS data.

2) In Figure 2, PIP₂ binding site should be presented in a close-up view and interacting residues should be described. Contact surface area should be calculated to show how extensive the interactions are (between two ENTH domains, between PIP₂ and ENTH alpha0, and between PIP₂ and ENTH No alpha0). If there is only a small contact area between PIP₂ and ENTH No alpha0, it could be merely an artifact of crystal packing.

3) The authors assume Thr104 of ENTH as the secondary PIP₂ binding site (Figure 3b). Native MS analysis with Thr104 mutant would support this idea.

4) In the middle of page 6, the authors state "As a further indication that the ENTH2 dimer, which sandwiched PIP₂, is functionally relevant, a similar dimer interface is observed in a crystal structure of the epsin Ent1 ENTH domain (ENTH1) from *S. cerevisiae* at 2.85 Å resolution (Figure 2c)." Just a similar crystal packing observed in both ENTH1 and ENTH2/PIP₂ does not support functional relevance. Furthermore, as ENTH1 dimerized without PIP₂, the dimerization of ENTH2 may not have been induced by PIP₂.

5) Hexameric ring model of ENTH (Figure 6c) is based on the ENTH2/PIP2 dimer crystal structure. As Thr104 is in the dimer interface, ENTH (at least ENTH No alpha0) cannot interact with ANTH which also requires Thr104 for interaction.

6) In page 8 line 7, the authors state "The insertions increase the surface complementarity between ENTH and the Hip1R ANTH domain." This should be described quantitatively. Anyway it is too speculative since the structure of ANTH/ENTH complex is not known at atomic resolution.

7) In the middle of page 8, the authors state "Based on *C. thermophilum* ANTH structure, ANTH domains of the Hip1R subfamily do not contain an N-terminal alpha helix. This is further confirmed by circular dichroism experiments performed on several ANTH and ENTH domains (Supplementary Figure 3)." It is not confirmed as CD signal is lost (Supplementary Figure 3c, 3e and 3f).

8) Figure 4 needs more detailed explanation.

9) In Supplementary Figure 5, only two data are shown in "f". Three data corresponding to "e" should be shown.

10) In the last sentence of the second paragraph of page 7, "In general, PIP2 dissociation constants are lower" should be "In general, PIP2 dissociation constants are higher".

Reviewer #3 (Remarks to the Author):

The manuscript by Garcia-Alai et al., describes the formation of complexes between the membrane-proximal domains of epsin (ENTH) and the ANTH domain of Hip1R, through PIP2 lipid interfaces. The authors use native mass spectrometry and other biophysical techniques to show how ENTH and ANTH domains assemble by sharing PIP2 molecules and how parts of this assembly process are conserved evolutionarily between yeast and human proteins. In addition the crystal structures of yeast ENTH2 and Hip 1 R-like ANTH domain of *C. thermophilum* are presented.

As requested by the associate editor, my comments focus on the mass spectrometry experiments presented in the manuscript. In general the paper is well written and the results are overall supported by the data and provide new insights into the assembly of ENTH and ANTH domains through PIP2 binding.

Main points.

Page 5, line 6: "Based on a direct MS approach". What was the approach taken for determining the dissociation constants? The reference given is for a review article by Erba & Zanobi. I would imagine that a titration method has been used, however, details would be very useful. Certainly in the supplementary section it would be good to see binding curves for these measurements. Were Hill type plots generated? Were the Kd values determined by linear regression? As much is made of these values, more detailed explanation is warranted.

Page 7, line16: "PIP2 dissociation constants are lower for Hip1R ANTH domains than ENTH..." perhaps true if you include the PICALM value, however, it is stated that this value is significantly different to the fungal protein domains. If you discount the value for human PICALM, then the ENTH KD values are lower than the Sla2 measurements.

Figure 4: Would definitely benefit from great annotation. Masses should be included on the spectra and the 6:6:19 complexes should be indicated. Actually supplementary figure 4b is much better, showing good resolution and revealing fine structural details. In some ways it is a shame it is tucked away in the supplemental section.

Figure 1a, +9 charge state of *C. thermophilum* ENTH appears to overlap with charge states from Cytc. Was this charge state excluded from further analysis?

Minor points.

Throughout the manuscript the authors jump between Hip1R and Sla2, or they state Hip1R related. This can be a little confusing. Consistency would be useful.

Page 3, line 3: define CALM.

Page 4, line 1: define GUV.

Figure 1a and all mass spectral figures, m/z is in the wrong place and should be in the middle below axis.

Page 7, line10: define PICALM

Page 10, line 2, "ENTH/ANTH complexes was also observed between murine..." The use of "was" makes it sound as if the experiment was carried out by the authors?

Reviewers' comments:

Reviewer #1 (Remarks to the Author):

This manuscript by Garcia-Alai is entitled Epsin and Hip1R form assemblies through phospholipid interfaces and yet the majority of the data, certainly the structural data rather than modelling concerns not the mammalian proteins as the title suggests but their fungal equivalents. This is inappropriate and seems to be an increasing trend on the yeast field to refer to mammalian proteins in the title instead of the fungal proteins on which the work is actually mainly done. The yeast data is good enough to stand on its own and the mammalian work is the weakest part of the paper as it is based largely on modelling studies and not compelling data. I think the title should be changed to reflect this. I think there would be a good case for removing the flakey sections on the mammalian proteins towards the end of the Results section but if the authors wish to retain it then they should back it up with further data. regardless of which route is chosen, more information and explanation and possibly further clarifying data is needed on the status of PIP2 native mass spec part

In general however, the paper is well written, the data presented well and the crystallography done properly and the native mass spec, which is a cornerstone of the work, is relatively new and interesting. There is a tendency to not discuss the mammalian literature sufficiently/selectively when it is referred to.

We appreciate the reviewer's concern that a clear distinction should be made between the data on the fungal proteins and the human system. We have changed the title of the manuscript and replaced human Hip1R with fungal Sla2. We would like to emphasize that the mammalian work is based on experimental evidence from native mass spectrometry and small angle X-ray scattering. We have shifted the focus towards the fungal system and added references to reflect the mammalian literature.

I have a serious technical issue concerning the use of PIP2

I also have a number of requests and questions that would need answering in a revised manuscript

My major technical concern focuses on the use of PIP2 in native mass spec. I can find no information as to whether it is in liposomes and indeed whether it is neat PIP2 or PIP2 mixed with the normal physiological other lipids. Indeed I am not even sure if it is PIP2 with long medium short or no (ie headgroup) acyl chains. The concentrations mentioned seem very large to me hence many sites would be saturated , which would not be in vivo. In the cell estimates of PIP2 concentration at plasma membrane are 1-5% but this does not take into account all that is bound by other proteins of both trafficking and especially signalling backgrounds. At the high concentrations used, does the PIP2 form aggregates, liposomes or membrane-like fragments. This needs to be characterised and I would suggest the experiments performed with relevant lipid mixtures

We realize that we should give more detailed information on the source of Phosphatidylinositol-4,5-bisphosphate diC8 (PIP2), which was obtained from Echelon. For the characterization of the PIP2 binding of ENTH and ANTH domains PI(4,5)P2 diC8 was used at sub-micellar concentration (60 μM ; CMC is approximately 200 μM) in an ammonium acetate solution. Native mass spectrometry is very sensitive in detecting micelles, and none were observed during the measurements. According to Gamper and Shapiro (J Physiol. 2007 582(Pt 3): 967–975), 60 μM PIP2 is in the same order of magnitude as in the physiological membrane environment.

The methods section was updated to clarify this technical issue (Page 18, 23 & 24), and it is more clearly defined the first time PIP2 is mentioned in the main text (Page 5). For the methods, the preparation of the complex is now described as follows (Page 23):

“Complexes were assembled after buffer exchange by mixing 10 μM ENTH, 10 μM ANTH and 60 μM PIP2 (Phosphatidylinositol-4,5-bisphosphate diC8 , Echelon). Thus, soluble PIP2 was used at a sub-micellar concentration that is in the range of its physiological concentration in the cell⁴⁵. “

And: on page 24:

“Samples were prepared by adding 60 μM PIP2 (Phosphatidylinositol-4,5-bisphosphate diC8 , Echelon) to the protein of interest (10 μM). For ENTH domains, cytochrome c from equine heart (PN 129021, Sigma-Aldrich) and for ANTH domains, carbonic anhydrase isoenzyme II from bovine erythrocytes (PN C2522, Sigma-Aldrich) was used as reference protein to test for unspecific clustering as described¹⁸. “

Throughout such as on page 4, but also in many other places, the authors do not state whether they are dealing with the fungal or human proteins. Many readers will be unsure of the origins of the proteins. This is important since there is a fundamental difference in endocytosis in yeast and mammals concerning many aspects including the need for actin and AP2 (the central controller in mammals not needed in yeast). Please make it clear by stating each time if it is a yeast or a human protein.

We have indeed struggled with the nomenclature for Sla2 and Hip1R, and we apologize for the confusion. We have now consistently introduced species-specific nomenclature, except when general statements are made on the structural features of the family. In this case, we have used the term Sla2/Hip1R.

P5&6 One decimal place for resolution is appropriate. There should be electron density shown for each structure of a sample piece and of key functional regions and ligands so that readers can judge whether or not the authors interpretations are appropriate. At 3.4Å a lot of things are unclear and often ambiguous. It is not acceptable to not show the electron density. A PDB check file could also be included.

We have now included a Supplementary figure (Figure S1) to show electron densities of the respective structures. We have rounded the low resolution structure to 3.4 Å and shown the region of the N-terminal helix of the ENTH2/PIP2 complex. We have also included PDB validation reports for the ENTH2/PIP2, Sla2 and ENTH1 structures.

P7 Why do the authors insist on calling CALM PICALM? PICALM is an invention of the alzheimers disease community – we all know it bind Phospholinosities. CALM has been published by several groups to have a 1-20uM affinity for PIP2 in membranes using a wide variety of well tried and tested techniques – what do the modelling studies show here? Is Mass spec a more reliable way of determining KDs? The conditions are certainly not physiological and it is well known that going from 100 to 250mM NaCl can move the Kd of epsin for PIP2 from 1uM to unmeasurably weak. Here 300mM NH4OAc is used and this could greatly affect affinity measurements. What are the apparent Kds nearer to physiological i.e. 160mM NH4OAc

We thank the reviewer to point out that CALM is the more commonly used nomenclature, and we have replaced PICALM with CALM.

Mass spectrometry is a reliable tool to characterize binding of small molecules to proteins without need for modeling. The method is especially useful when the number of binding sites has to be assessed or weak interactions up to millimolar K_D s are studied. Biophysical techniques such as isothermal titration calorimetry (ITC) and surface plasmon resonance (SPR) do not distinguish multiple binding sites, unless their K_D s are very dissimilar. These are bulk techniques, whereas native mass spectrometry is unique in that it can distinguish binding site occupancies by mass difference. This is nicely illustrated by the measurement on human ENTH1 in complex with PIP2. Bulk techniques such as SPR or ITC would consider the binding stoichiometry as 1:1 between ENTH and PIP2, but native MS unmasked the presence of a singular hexameric species, which has 6 copies of ENTH and 6 copies of PIP2, maintaining a 1:1 stoichiometry.

Upon suggestion from the reviewer, measurements were repeated at 160 mM NH_4OAc for ENTH1 and ANTH (both *S. cerevisiae*) and CALM ANTH (*H. sapiens*) in triplicates and the obtained K_D s reveal no major deviation from the measurements in 300 mM NH_4OAc . The results were added to the Supplementary Table 1 and mentioned in the table caption:

“Unless stated otherwise, measurements were performed in 300 mM NH_4OAc pH 8.0. Samples marked with an asterisk were measured at the more physiological concentration of 160 mM NH_4OAc pH8.0. ”

Main text page 5:

“Results were comparable for the MS optimized ammonium acetate concentration of 300 mM and the more physiological ionic strength of 160 mM ammonium acetate (Supplementary Table S1). ”

P8 In order to confirm whether an alpha helix0 exists in HIPR-like ANTH domains, it would be necessary to carry our CD \pm liposomes and I can see no indication this was done. Also does the Nterminus predict as an amphipathic helix – use Bruno Antonny’s server (heliquest).

The Heliquest server does not predict an amphipathic helix for any of the sequences of Sla2 or Hip1R presented in Supplementary Figure S2. Moreover, as this figure indicates, the N-terminus of Hip1R-like ANTH domains vary extensively, indicating there is no evolutionary conserved feature, which would be expected if an N-terminal amphipathic helix were present.

We have now performed CD spectra for human CALM ANTH and Hip1R ANTH in the presence of submicellar and micellar concentrations of the detergent n-Dodecyl-beta-Maltoside (DDM) in addition to PIP2. This provides an amphipathic environment without the need of working with liposomes. Detergents have been extensively used in CD experiments for induction and stabilization of helical structures in proteins (Zhong et al, 1992).

Supplementary Figure 3g shows the CD spectra for the human CALM ANTH domain. There is a clear increase in signal (both minima at 208nm and 220 nm, indicative of alpha-helical content) in the presence of DDM and PIP2 (pale blue line) when compared to the protein in buffer alone or with PIP2. In addition, when only the detergent above micellar concentrations is added; we further observe an increase in the signal. This experiment shows that it is actually the presence of the micellar environment and not only the phospholipid *per-se* that is inducing/stabilizing helical structures in the protein and is in agreement with what has been previously reported for the ANTH domain from CALM that contains an $\alpha 0$ amphipathic helix (Miller et al., 2015). In contrast, when we repeat this same experiment with Hip1R ANTH (Supplementary figure 3h) we only observe loss of signal. The protein seems to undergo unfolding in the presence of an amphipathic environment, which would not be expected for a protein that would contain a lipid-binding inducible alpha helix.

Reference:

Zhong, L., and Johnson, W. C., Jr. (1992) Environment affects amino acid preference for secondary structure, Proc. Natl. Acad. Sci. U.S.A. 89, 4462–4465.

We have inserted the following text into the main manuscript (Page 9):

*“Sla2/Hip1R ANTH domains display a loss of secondary structure or aggregation at high PIP2 concentrations (Supplementary Figure 3 c and f). To address whether this is caused by PIP2 itself, or the presence of an amphipathic environment we repeated the circular dichroism measurements in the presence of submicellar and micellar concentrations of the detergent n-Dodecyl-beta-Maltoside (DDM) with and without PIP2 (Supplementary Figure 3 g and h). Human Hip1R ANTH domain shows a loss of signal with DDM (Supplementary Figure 3h). The protein seems to undergo unfolding in the presence of an amphipathic environment, which would not be expected for a protein that contains a helix inducible by lipid binding. In contrast, CALM displays a clear increase in signal in the presence of DDM and PIP2 compared to the protein in buffer alone or with PIP2. In addition, when only the detergent above micellar concentrations is added, we further observe an increase in the signal. This experiment shows that it is actually the presence of the micellar environment and not only the phospholipid *per-se* that is inducing/stabilizing helical structures in the protein and is in agreement with what has been previously reported for CALM that contains an $\alpha 0$ amphipathic helix⁹.”*

P9, removal of an ANTH occurs due to collision in MS as I understand it. Do more come off with more time? How relevant do the authors consider this and why since collisions don't occur in the cell.

Collision-induced dissociation (CID) mass spectrometry is a technical tool that can be used to obtain more information on the protein complex topology. This process is not comparable to dissociation as it occurs in solution. Multiple collisions with gas atoms result in gradual unfolding and charge migration of

proteins until they finally become ejected from the complex. The number of dissociating proteins is limited by the number of charges that remain on the residual complexes. Since the first dissociation event takes a vast number of charges (in literature often referred to as asymmetric charge partitioning), additional dissociations are less likely to occur. In general, small proteins at the periphery of a complex require a lower degree of collision energy for dissociation. Since we only observe the dissociation of the larger ANTH domain during collision experiments, we conclude that the ANTH domains are on the periphery. The ENTH domains are protected from dissociation by collision, and form the core of the complex. The assembly pathway of the human complexes, where ANTH domains bind to preformed PIP2-containing ENTH hexamers strongly supports this hypothesis.

We have added further explanation to Page 11:

“Further information on complex topology was obtained in collision-induced dissociation (CID) MS, in which proteins unfold partially until small proteins from the periphery are ejected. Here, hetero 12-mers and 16-mers show identical dissociation pathways. The larger ANTH domain is ejected from the complexes while all PIP2 and ENTH molecules remain bound to the complexes, indicating localization of ANTH in the periphery around an ENTH/PIP2 core (Figure 3a and Supplementary Figure 4).”

P10 Why is full length and not ANTH HIPR used?

We thank the reviewer for pointing this out. Indeed this is referring to another experiment in the literature. We have exclusively used the ANTH domain from Hip1R and have modified the text to clarify this issue (Page 11):

“It was previously reported that full length Hip1R can only be pulled down by murine epsin ENTH in the presence of PIP2, suggesting that the PIP2-dependent formation of ENTH/ANTH complexes could be conserved in mammals¹³. To verify evolutionary conservation of ENTH/ANTH/PIP2 complexes, a mixture of human epsin-1 ENTH domain with human Hip1R ANTH domain and PIP2 was analyzed by native MS, showing less homogeneity with a larger variety of oligomers (Figure 5b).”

P10 The mammalian protein oligomers are extremely heterogeneous – why should the 6 ENTH based complexes be considered to be biologically relevant? What fraction actually has a hexamer ENTH at its core?

In general the data on mammalian proteins seems rather preliminary and needs firming up.

The reviewer is correct that mammalian oligomers exhibit a large heterogeneity when it comes to the number of Hip1R ANTH (0-8). However, all oligomers contain an ENTH core, and both native MS and SAXS show that the human ENTH/PIP2 complex is extremely homogeneous with practically 95 % of the protein forming a hexamer. This is in contrast with fungal ENTH, where the bulk SAXS technique indicates the sample is heterogenous (data not included), but native MS can be used to detect individual ENTH domains and identify two PIP2 binding sites.

In fact, this divide-and-conquer approach to analyse different aspects of the assembly of the ENTH/ANTH/PIP2 complex in yeast and human is central to the paper. The human ENTH/PIP2 complex

forms a hexamer, which shows it can form a core that is then decorated by ANTH and possibly additional ENTH molecules.

In mammalian cells, as opposed to fungal cells, the roles of epsin and HIPR are proposed to be very different. Hip1R is meant to interact with clathrin light chains distant from the membrane surface and epsin has been extensively referenced as causing membrane curvature and more recently, scission, as deletion of all epsins results in the production of bunches of grapes (McMahon lab). How does the formation of multimers of some size in mammals fit with these models. The authors should explain these points and how their speculative model fits into the wealth of existing data.

Indeed, our native MS experiments indicate that there is a fundamental difference in the assembly of epsin and Sla2/Hip1R through PIP2 between yeast and human. Human epsin ENTH forms a hexameric scaffold in the presence of PIP2, and independent of the presence of Hip1R. According to the SAXS molecular envelope, this hexamer could insert into the membrane, and further PIP2-driven oligomerization could occur. We thank the reviewer for pointing out the work by McMahon showing that mammalian epsin plays a role in fission. In particular, the proposal that epsin could assist dynamin or replace it altogether as the vesicle undergoes scission is very appealing. PIP2-driven oligomerization of epsin will contribute to the closure of the neck, and the model presented in Boucrot et al (2012) requires a mechanism for epsin-assembly. Our current work could provide such a mechanism. We have now added a paragraph at the end of the discussion to better emphasize the possible implications of PIP2-driven scaffolding of epsins, and we also highlight throughout the text that the assembly of epsin with or without Hip1R provides a combined mechanism of helix insertion into the membrane and scaffolding.

Reviewer #2 (Remarks to the Author):

The authors previously reported that yeast clathrin adaptor proteins, Sla2 and Ent1, co-assemble in a PIP2-dependent manner (Skruzny 2015). In this paper, the authors report biophysical analysis of this interaction. Although extensively studied, there are many concerns to be addressed.

1) The authors claim that the ENTH and ANTH domains both bind two PIP2 by native MS analysis (Figures. 1 and 3). It is desirable to show ITC data to support the native MS data.

The ITC experiment for the interaction between rat epsin and PIP2 was published by Ford et al. in 2002. The reported K_D is around 1 mM. We have tried repeating this experiment obtaining similar values but the quality of the data is not good enough to calculate any stoichiometry. In general, for this low affinity interactions (especially for lipid binding proteins) ITC is not the most suitable technique. ITC is well suited to determine K_D s, however providing clues about binding stoichiometry or affinity of weak binders are more difficult to obtain. Native MS has proven useful for stoichiometry determination and affinity of weak binders. Multiple studies have shown that data is comparable to other methods (Erba & Zenobi 2011 and Sun et al. 2006). Furthermore, we have analyzed lipid binding in different conditions by native MS and obtained comparable results supporting robustness of the approach (page 5, Supplementary Table 1).

2) In Figure 2, PIP2 binding site should be presented in a close-up view and interacting residues should be described. Contact surface area should be calculated to show how extensive the interactions are (between two ENTH domains, between PIP2 and ENTH alpha0, and between PIP2 and ENTH No alpha0). If there is only a small contact area between PIP2 and ENTH No alpha0, it could be merely an artifact of crystal packing.

We have now included a close up view of the residues surrounding the PIP2 binding site (Supplementary Figure S1). We used the PISA server to analyze the contact surfaces between the ENTH2 and PIP2 molecules, which has led to the following addition (Page 6):

“To analyze the contribution of the PIP2 molecule to complex formation, we calculated the buried surface area contributions for each molecule with PISA²⁴. The total solvent accessible area of the phosphatidylinositol head group of PIP2 is 488 Å². Most of the head group is buried by the ENTH2_{α0} molecule (306 Å², or 63 % of the total available surface area of the PIP2 molecule), whereas the ENTH2_{Noα0} molecule covers 95 Å², or 20 % of the available surface area). The buried solvent area between ENTH2_{α0} and ENTH2_{Noα0} is 1820 Å², with almost equal contributions from ENTH2_{α0} (955 Å²) and ENTH2_{Noα0} (865 Å²). However, most of the buried area (1020 Å²) contributed to the dimer interface comes from the α0 helix of the ENTH_{α0} domain, which is repositioned by the PIP2 molecule to facilitate dimer formation. The effect of PIP2 binding is therefore twofold; it attaches the ENTH domain to the membrane and displaces the α0 helix so it can form a dimer with a second ENTH domain.”

3) The authors assume Thr104 of ENTH as the secondary PIP2 binding site (Figure 3b). Native MS analysis with Thr104 mutant would support this idea.

We thank the reviewer for this suggestion and prepared a Thr104Glu mutant of *S. cerevisiae* ENTH1. PIP2 binding and ENTH/ANTH/PIP2 complex formation was studied with native MS. The choice to replace threonine with glutamate is based on previous mutagenesis studies *in vivo* (Skruzny et al. 2015).

Figure 1b and Supplementary Table 1 have been updated and the Thr104Glu mutant was added to the analysis. It clearly shows a reduced PIP2 binding compared to the wild type. After correction for unspecific binding, the species with 2 bound PIP2 molecules is less than 3% of the unbound protein signal. As PIP2 binding to the other binding site (that was not mutated) is also impaired, it can be assumed that the suggested allosteric effect is also impaired in the Thr104Glu mutant similar to the Sla2 ANTH mutant.

No complexes of ENTH/ANTH/PIP2 could be formed with the ENTH1 Thr104Glu mutant. The new results demonstrate the importance of Thr104 for the binding of PIP2 and also for the formation of ENTH/ANTH/PIP2 complexes. A supplementary figure was prepared (Figure S6) and the results regarding PIP2 binding to ENTH are discussed in the main text (page 6):

*“Native MS on a Thr104Glu mutant of ENTH1 from *S. cerevisiae* shows a reduction of PIP2 binding, where the binding of two PIP2 molecules is barely observed (Figure 1b) indicating impairment of the binding site.*

and the lack of complex formation is discussed on page 10:

“Furthermore, complex formation was impaired for the ENTH1 Thr104 mutant (Supplementary Figure 6), confirming previous ITC measurements¹⁴. “

4) In the middle of page 6, the authors state “As a further indication that the ENTH2 dimer, which sandwiched PIP2, is functionally relevant, a similar dimer interface is observed in a crystal structure of the epsin Ent1 ENTH domain (ENTH1) from *S. cerevisiae* at 2.85 Å resolution (Figure 2c).” Just a similar crystal packing observed in both ENTH1 and ENTH2/PIP2 does not support functional relevance. Furthermore, as ENTH1 dimerized without PIP2, the dimerization of ENTH2 may not have been induced by PIP2.

The main driving force in the dimerization of the ENTH domain is the repositioning of the $\alpha 0$ helix. In the crystal structure of epsin Ent1 ENTH domain, an MPD molecule is situated close to the $\alpha 0$ helix. This has led to a displacement of the $\alpha 0$ helix, so that an ENTH/ENTH dimer could form. We have also calculated the buried surface area (BSA) for the ENTH1 dimer interface that is relevant for the comparison with the ENTH2/PIP2 interface. The BSA is somewhat lower (1750 vs 1820 Å²), but once again the contribution from the $\alpha 0$ helix is dominant, covering 900 Å² (about 50 %) of the total buried surface area. To clarify this, we have added the following sentence (Page 7):

“Here, a 2-methyl-2,4-pentanediol (MPD) molecule from the crystallization solution has caused a displacement of the $\alpha 0$ helix to allow dimer formation between ENTH1 domains.”

5) Hexameric ring model of ENTH (Figure 6c) is based on the ENTH2/PIP2 dimer crystal structure. As Thr104 is in the dimer interface, ENTH (at least ENTH No alpha0) cannot interact with ANTH which also requires Thr104 for interaction.

The orientation of the ENTH dimer within the hexameric rings is such that one Thr104 interface is involved in dimerization, while the second Thr104 interface is pointing outward, ready to form further interactions with either an ANTH domain or an ENTH domain through a PIP2 interface. To clarify this better, we have modified Figure 6c to highlight not only the Thr104 interfaces that are available for ENTH binding, but also the 3 interfaces on the surface of the hexamer, ready to form additional interactions with ENTH or ANTH domains. We have also added this sentence (Page 12):

“Similarly, the interface involving Thr104 is alternatively used for hexamer formation as well as further oligomerization through phospholipid interfaces.”

And we have added the following sentence to the legend of Figure 6:

“The Thr104 residues involved in ENTH/ENTH homodimerization are colored pink, whereas the Thr104 residues present on the surface of the hexamer are colored green.”

6) In page 8 line 7, the authors state “The insertions increase the surface complementarity between ENTH and the Hip1R ANTH domain.” This should be described quantitatively. Anyway it is too speculative since the structure of ANTH/ENTH complex is not known at atomic resolution.

Indeed we agree with the reviewer that the surface complementarity between ENTH and Hip1R ANTH cannot be quantified with the 13 Å EM map. We have deleted this sentence.

7) In the middle of page 8, the authors state “Based on *C. thermophilum* ANTH structure, ANTH domains of the HiP1R subfamily do not contain an N-terminal alpha helix. This is further confirmed by circular dichroism experiments performed on several ANTH and ENTH domains (Supplementary Figure 3).” It is not confirmed as CD signal is lost (Supplementary Figure 3c, 3e and 3f).

We agree with the Referee that the loss of the CD signal cannot confirm the absence of α 0 helix, neither the loss of helical structure. The text was misleading and is now reformulated and additional experiments were performed with DDM detergent to show that it is the amphipathic environment that causes the loss of CD signal for Sla2/Hip1R. On page 9, we have added:

“To address whether this is caused by PIP2 itself, or the presence of an amphipathic environment we repeated the circular dichroism measurements in the presence of submicellar and micellar concentrations of the detergent n-Dodecyl-beta-Maltoside (DDM) with and without PIP2 (Supplementary Figure 3 g and h). Human Hip1R ANTH domain shows a loss of signal with DDM (Supplementary Figure 3h). The protein seems to undergo unfolding in the presence of an amphipathic environment, which would not be expected for a protein that contains a helix inducible by lipid binding. In contrast, CALM displays a clear increase in signal in the presence of DDM and PIP2 compared to the protein in buffer alone or with PIP2. In addition, when only the detergent above micellar concentrations is added, we further observe an increase in the signal. This experiment shows that it is actually the presence of the micellar environment and not only the phospholipid per-se that is inducing/stabilizing helical structures in the protein and is in agreement with what has been previously reported for CALM that contains an α 0 amphipathic helix⁹.”

8) Figure 4 needs more detailed explanation.

We have added cartoons of the ENTH (blue) and ANTH (green) assembly measured for the three configurations, as well as the MS/MS spectrum on the left (ANTH alone) and the remaining complex (on the right). The figure legend has been extended to better describe the most prominent species detected with the MS and MS/MS spectra.

9) In Supplementary Figure 5, only two data are shown in “f”. Three data corresponding to “e” should be shown.

The other curves were hidden underneath. Color and thickness have been modified to visualize the individual curves.

10) In the last sentence of the second paragraph of page 7, “In general, PIP2 dissociation constants are lower” should be “In general, PIP2 dissociation constants are higher”.

Correct, the phrasing was changed to: “In general, PIP2 dissociation constants are higher”.

Reviewer #3 (Remarks to the Author):

The manuscript by Garcia-Alai et al., describes the formation of complexes between the membrane-proximal domains of epsin (ENTH) and the ANTH domain of Hip1R, through PIP2 lipid interfaces. The authors use native mass spectrometry and other biophysical techniques to show how ENTH and ANTH domains assemble by sharing PIP2 molecules and how parts of this assembly process are conserved evolutionarily between yeast and human proteins. In addition the crystal structures of yeast ENTH2 and Hip 1 R-like ANTH domain of *C. thermophilum* are presented.

As requested by the associate editor, my comments focus on the mass spectrometry experiments presented in the manuscript. In general the paper is well written and the results are overall supported by the data and provide new insights into the assembly of ENTH and ANTH domains through PIP2 binding.

Main points.

Page 5, line 6: “Based on a direct MS approach”. What was the approach taken for determining the dissociation constants? The reference given is for a review article by Erba & Zanobi. I would imagine that a titration method has been used, however, details would be very useful. Certainly in the supplementary section it would be good to see binding curves for these measurements. Were Hill type plots generated? Were the K_d values determined by linear regression? As much is made of these values, more detailed explanation is warranted.

“Direct MS approach” refers to a single point measurement that was introduced by Sun et al 2006. The mentioned review article was supplemented by the primary reference. Due to the number of different proteins investigated titrations were not performed. We agree that this is the most reliable method to determine K_D s. However, since identical conditions were employed the results are comparable revealing allostery between the two binding sites in both proteins and the distinct behavior of CALM ANTH. This is further supported by comparable results obtained at lower ionic strength.

Page 7, line16: “PIP2 dissociation constants are lower for Hip1R ANTH domains than ENTH...” perhaps true if you include the PICALM value, however, it is stated that this value is significantly different to the fungal protein domains. If you discount the value for human PICALM, then the ENTH K_D values are lower than the Sla2 measurements.

Thanks to these remarks the phrasing was changed to: “*In general, PIP2 dissociation constants are higher*”

Figure 4: Would definitely benefit from great annotation. Masses should be included on the spectra and the 6:6:19 complexes should be indicated. Actually supplementary figure 4b is much better, showing good resolution and revealing fine structural details. In some ways it is a shame it is tucked away in the supplemental section.

Additional annotations were added to Figure 4 and the figure legend was updated to better support the findings:

*Figure 4. ENTH/ANTH/PIP2 complex formation in fungi. 6:6:~18 and 8:8:~24 ENTH/ANTH/PIP2 complexes are the observed stoichiometries in native MS measurements. A cartoon of the most prominent complex is shown with ANTH in green and ENTH in blue. Complexes from *C. thermophilum* (green) and *S. cerevisiae*, containing ENTH1 (orange) or ENTH2 (red), show the same stoichiometries, signal ratios and dissociation pathways in collision induced dissociation (CID) MS/MS. Here, the dissociation of the +40 charged 8:8:25±3 ENTH1/ANTH Sla2/PIP2 complex into partially unfolded ANTH Sla2 (top left, cartoon shows green ANTH domain) and a residual 8:7:23±1 complex (top right, showing also a cartoon of the remaining complex) is depicted. The annotation shows the stoichiometry (ENTH:ANTH:average PIP2 number), charge state of the main peaks, and average experimental masses. Ranges of PIP2 numbers, statistical errors and an average FWHM value rating the MS resolution of all complexes can be found in Supplementary Table 3. ANTH dissociation in CID MS/MS measurements of 6:6:18±1 ENTH2/ANTH Sla2/PIP2 complexes from *S. cerevisiae* is presented in Supplementary Figure 4b.*

The important aspect is the identical dissociation behavior revealing the subunit organization in the main complex. We therefore decided not to move the spectrum of the lower abundant 6:6 species from the supplement. However, the high quality is further supporting that both species have the same arrangement and that the lipids are associated with the ENTH complex core.

Figure 1a, +9 charge state of *C. thermophilum* ENTH appears to overlap with charge states from Cytc. Was this charge state excluded from further analysis?

The referee is correct, the overlap causes difficulties in analysis and the approach to solve this issue was only reported for other proteins in the text. This updated, more general explanation can now be found in the methods section:

“If there was signal overlap of the ENTH or ANTH domain with signal of the PIP2-bound reference protein, this charge state was excluded for the calculation of the ratio of unspecific binding. As an approximation the unspecific binding that was determined using the remaining charge states of the reference protein was subtracted from overlapping signals and the residual signals of the overlapping peaks were considered for further analysis.”

Minor points.

Throughout the manuscript the authors jump between Hip1R and Sla2, or they state Hip1R related. This can be a little confusing. Consistency would be useful.

Page 3, line 3: define CALM.

Page 4, line 1: define GUV.

Figure 1a and all mass spectral figures, m/z is in the wrong place and should be in the middle below axis. Page 7, line 10: define PICALM

Page 10, line 2, “ENTH/ANTH complexes was also observed between murine...” The use of “was” makes it sound as if the experiment was carried out by the authors?

All minor points have now been addressed.

REVIEWERS' COMMENTS:

Reviewer #1 (Remarks to the Author):

This paper has been much improved by the changes and additions made to both experiments and text.

Reviewer #2 (Remarks to the Author):

I am satisfied with the revised manuscript. I have only four minor comments.

- 1) Please specify cyan and pale blue structures in Figure 3b legend. There is no "blue" structure in Figure 3b.
- 2) In Supplementary Figure 1 legend, 2M_{Fo}-DF_c should be 2m_{Fo}-DF_c.
- 3) In Supplementary Figure 3g and 3h, red and orange are difficult to distinguish.
- 4) Line colors of Supplementary Figure 5e and 5f seem to be mixed up.

Reviewer #3 (Remarks to the Author):

The authors have responded to my initial review of the manuscript and have adequately answered/modified the paper accordingly.

For the revision, only reviewer 2 required additional modifications to the manuscript, as listed below.

Reviewer #2 (Remarks to the Author):

I am satisfied with the revised manuscript. I have only four minor comments.

1) Please specify cyan and pale blue structures in Figure 3b legend. There is no 1C;blue 1D; structure in Figure 3b.

We have modified the Figure legend of Figure 3b as follows:

b, Composite model of the ENTH and ANTH_{Sla2} complex based on the low resolution EM structure (original ANTH model, blue)¹⁴ and the high resolution X-ray crystal structures presented here (Sla2, violet; ENTH2/PIP2, yellow).

Has been replaced by

b, Composite model of the ENTH and ANTH_{Sla2} complex based on the low resolution EM structure¹⁴ and the high resolution X-ray crystal structures presented here (ENTH2/PIP2, yellow, and superimposed ANTH domains of Sla2 (violet), AP180 (cyan) and CALM (pale cyan).

2) In Supplementary Figure 1 legend, 2M_{Fo}-D_{Fc} should be 2m_{Fo}-D_{Fc}.

Corrected

3) In Supplementary Figure 3g and 3h, red and orange are difficult to distinguish.

Lines have been thickened and are now differently colored

4) Line colors of Supplementary Figure 5e and 5f seem to be mixed up.

Fixed.

End of reviewer comments.